# Weisfeiler and Leman Go Gambling: Why Expressive Lottery Tickets Win

**Lorenz Kummer** [1 2]  **Samir Moustafa** [1 2]  **Anatol Ehrlich** [1 2]  **Franka Bause** [1 2]
**Nikolaus Suess** [1]  **Wilfried N. Gansterer** [1]  **Nils M. Kriege** [1 3]

## Abstract

The lottery ticket hypothesis (LTH) is well-studied for convolutional neural networks but has been validated only empirically for graph neural networks (GNNs), for which theoretical findings are largely lacking. In this paper, we identify the expressivity of sparse subnetworks, i.e. their ability to distinguish non-isomorphic graphs, as crucial for finding winning tickets that preserve the predictive performance. We establish conditions under which the expressivity of a sparsely initialized GNN matches that of the full network, particularly when compared to the Weisfeiler-Leman test, and in that context put forward and prove a *Strong Expressive Lottery Ticket Hypothesis*. We subsequently show that an increased expressivity in the initialization potentially accelerates model convergence and improves generalization. Our findings establish novel theoretical foundations for both LTH and GNN research, highlighting the importance of maintaining expressivity in sparsely initialized GNNs. We illustrate our results using examples from drug discovery.

## 1. Introduction

Graph Neural Networks (GNNs) have emerged as powerful tools for learning over graph-structured data. Complex objects such as molecules, proteins or social networks are represented as graphs. Nodes represent parts and edges their relations, both can be enriched with features. GNNs generalize established deep learning techniques and extend their applicability to new domains such as financial and social network analysis, medical data analysis, or chem- and bioinformatics (Lu & Uddin, 2021; Cheung & Moura, 2020; Sun et al., 2021; Gao et al., 2022; Wu et al., 2018; Xiong et al., 2021). One of the key fields in GNN research is expressivity, that is, a GNN's ability to distinguish non-isomorphic graphs. This expressivity is often benchmarked using the Weisfeiler-Leman (WL) test, with the 1-WL test being a standard baseline for many models. Key contributions to this field include Deep Sets (Zaheer et al., 2017), which introduced permutation-invariant learning on sets, and the Graph Isomorphism Network (GIN) (Xu et al., 2019), which explicitly aimed to match the expressivity of the 1-WL test. GIN set a new standard in graph representation learning by using a simple yet powerful aggregation function that ensures the model's ability to distinguish non-isomorphic graphs at a level comparable to the 1-WL test. The combination of these advances has shaped the current understanding of GNN expressivity, where models are judged by their capacity to approximate or exceed the performance of 1-WL in distinguishing complex graph structures. A large amount of research has been devoted to improving the expressivity of GNNs, aiming to go beyond the limitations of 1-WL and increase the model's capacity to distinguish between structurally distinct graphs (Morris et al., 2023). Despite these efforts, the 1-WL test continues to distinguish most graphs in widely used benchmarks (Morris et al., 2021; Zopf, 2022), demonstrating that, in many scenarios, it is not necessary to go beyond this foundational approach.

Practically, in drug discovery and molecular property prediction, the impact of model expressivity on prediction quality is not entirely clear. Structurally similar molecules can exhibit drastically different potencies towards the same protein target, a phenomenon known as activity cliffs (Pérez-Villanueva et al., 2015). The accurate identification of these activity cliffs is a challenging but essential task, as misclassifications may result in a drug candidate being erroneously interpreted as safe or toxic. Furthermore, distinguishing between stereoisomers is crucial, as these are molecules with the same molecular formula and bond structure but differ only in the three-dimensional spatial arrangement of their atoms. Even such small changes can significantly influence a molecule's biological activity and pharmacological profile.

The Lottery Ticket Hypothesis (LTH), originally proposed by Frankle & Carbin (2018), introduced the idea that large, randomly initialized neural networks contain smaller, train-

[1]Faculty of Computer Science, University of Vienna, Vienna, Austria [2]Doctoral School Computer Science, University of Vienna, Vienna, Austria [3]Research Network Data Science, University of Vienna, Vienna, Austria. Correspondence to: Lorenz Kummer <lorenz.kummer@univie.ac.at>.

*Proceedings of the $42^{nd}$ International Conference on Machine Learning*, Vancouver, Canada. PMLR 267, 2025. Copyright 2025 by the author(s).

able subnetworks, or *winning tickets*, that can match the performance of the full network. This hypothesis has gained considerable traction, especially in the context of deep learning, where iterative pruning techniques identify these subnetworks, significantly reducing model size and computational cost while preserving performance. LTH has since been extended to various domains, including GNNs. In GNNs, LTH has been explored primarily through the pruning of both graph structures (e.g., adjacency matrices) and model parameters (e.g., weights), leading to the discovery of Graph Lottery Tickets (GLTs), which maintain model performance while substantially reducing computational overhead (Chen et al., 2021a; Tsitsulin & Perozzi, 2023). Liu et al. (2024) provide a comprehensive survey on LTH and related works.

Despite extensive work on LTH in GNNs, and analogous investigations in related domains having yielded fundamental insights (Kummer et al., 2025), the link between LTH and GNN expressivity remains unexplored. While prior research focuses on efficiency, the effect of pre-training pruning on preserving expressivity is largely overlooked. Understanding this connection could reveal how sparse initialization impacts a GNN's ability to distinguish complex graph structures, and addressing this gap could advance more efficient graph learning models that maintain expressivity. Our work contributes to closing this gap by providing both formal and empirical evidence that preserving expressivity in sparsely initialized GNNs is crucial for finding winning tickets.

## 1.1. Related Work

LTH posits that large, randomly initialized neural networks contain smaller subnetworks, or winning tickets, that can match the full network's performance when trained in isolation (Frankle & Carbin, 2018). Through iterative pruning, these subnetworks are identified, significantly reducing the parameter count while preserving performance. Frankle et al. (2019) enhance LTH by introducing Iterative Magnitude Pruning (IMP) with rewinding, pruning subnetworks early in training rather than at initialization, thus finding sparse winning tickets in deep neural networks (DNNs) like ResNet-50, maintaining both accuracy and stability. Malach et al. (2020) extend LTH by proving the Strong Lottery Ticket Hypothesis (SLTH), showing over-parameterized networks contain subnetworks that achieve high accuracy without training, with theoretical guarantees for deep and shallow networks. Zhang et al. (2021a) theoretically explain LTH's improved generalization, showing pruned networks enlarge the convex region in the optimization landscape, enabling faster convergence and fewer samples for zero generalization error. Additionally, Zhang et al. (2021b) use Inertial Manifold Theory to validate LTH, showing that pruned subnetworks match dense network performance without repeated pruning and retraining. Finally, da Cunha et al. (2022) prove SLTH for CNNs, showing that large, randomly

initialized CNNs can be pruned into subnetworks approximating fully trained models. Zhang et al. (2019) propose Eager Pruning, an LTH-based method that prunes DNNs early in training, reducing computation without accuracy loss. Their efficient hardware architecture achieves significant speedups and energy efficiency over Nvidia GPUs, suiting energy-constrained applications. Chen et al. (2021b) embed ownership verification into sparse subnetworks via graph-based signatures, resilient to fine-tuning and pruning attacks, enabling verification without performance impact.

In GNNs, LTH approaches typically treat pruning the graph and the GNN's trainable parameters as a unified task. Specifically, Chen et al. (2021a) introduce the Unified GNN Sparsification (UGS) framework, which prunes both graph adjacency matrices and model weights to identify GLTs and sparse subnetworks that maintain performance while reducing computational cost. Wang et al. (2022) propose transforming random subgraphs and subnetworks into GLTs through hierarchical sparsification and regularization-based pruning, achieving high performance with substantial sparsity. Tsitsulin & Perozzi (2023) present the GLT Hypothesis, suggesting that any graph contains a sparse substructure capable of preserving the performance of graph learning algorithms. Efficient algorithms are developed to find these GLTs, demonstrating that GNN performance is retained even on sparse subgraphs. Sui et al. (2023) introduce a co-pruning framework that prunes input graphs and model weights in both inductive and transductive settings, identifying GLTs while maintaining performance at high sparsity levels. Hui et al. (2023) improve the UGS framework by introducing an auxiliary loss for better edge pruning and a min-max optimization for robustness under high sparsity, enhancing pruning effectiveness. Zhang et al. (2024) present an automated adaptive pruning framework to identify GLTs in GNNs, optimizing graph and GNN sparsity without manual intervention and improving scalability in deeper GNNs. Yuxin et al. (2024) introduce a scalable graph structure learning method based on LTH, which prunes adjacency matrices and model weights to maintain performance under adversarial conditions. Finally, Yan et al. (2024) propose Multicoated Supermasks (M-Sup) and folding techniques to optimize GNNs based on SLTH, enhancing memory efficiency while maintaining performance.

## 1.2. Contribution

We formally link GNN expressivity to LTH by establishing criteria that pruning mechanisms—both graph and parameter pruning—must satisfy to preserve prediction quality. We demonstrate the existence of trainable subnetworks within moment-based GNNs that match 1-WL expressivity, putting forward the Strong Expressive Lottery Ticket Hypothesis (SELTH) as a novel, GNN-specific extension of the classical SLTH. We subsequently argue that critical computational

paths (i.e., those whose removal degrades performance) are subsets of these maximally expressive subnetworks. Furthermore, we show that expressive sparse initializations can improve generalization and convergence. We also identify cases where expressivity loss is irrecoverable, exemplified by molecular property prediction in a medical context. Finally, we empirically confirm that GNN parameter pruning impacts post-training accuracy and that more expressive sparse initializations are more likely to be winning tickets.

## 2. Preliminaries

A *graph* $G$ is a pair $(V, E)$ of a finite set of *nodes* $V$ and *edges* $E \subseteq \{\{u, v\} \subseteq V\}$. The set of nodes and edges of $G$ is denoted by $V(G)$ and $E(G)$, respectively. The *neighborhood* of a node $v$ in $V(G)$ is $N(v) = \{u \in V(G) \mid \{u, v\} \in E(G)\}$. If a bijection $\varphi \colon V(G) \to V(H)$ with $\{u, v\} \in E(G) \iff \{\varphi(u), \varphi(v)\} \in E(H)$ for all $u, v \in V(G)$ exists, we call the two graphs $G$ and $H$ *isomorphic* and write $G \simeq H$. For two graphs with designated roots $r \in V(G)$ and $r' \in V(H)$, the bijection must further satisfy $\varphi(r) = r'$. The equivalence classes induced by $\simeq$ are referred to as *isomorphism types*. A function $l \colon V(G) \to \Sigma$ with an arbitrary codomain $\Sigma$ is called a *node coloring*. Then, a *node colored* or *labeled graph* $(G, l)$ is a graph $G$ endowed with a node coloring $l$. We call $l(v)$ a *label* or *color* of $v \in V(G)$. For labeled graphs, the bijection $\varphi$ must additionally satisfy $l(v) = l(v')$ if $\varphi(v) = v'$ for all $v \in V(G)$. We denote a multiset by $\{\!\{\ldots\}\!\}$.

**The Weisfeiler-Leman algorithm.** Let $(G, l)$ denote a labeled graph. In every iteration $t > 0$, a node coloring $c_l^{(t)} \colon V(G) \to \Sigma$ (where the subscript indicates the coloring is based on the initial labeling function $l$) is computed, which depends on the coloring $c_l^{(t-1)}$ of the previous iteration. At the beginning, the coloring is initialized as $c_l^{(0)} = l$. In subsequent iterations $t > 0$, the coloring is updated according to $c_l^{(t)}(v) = \text{HASH}\left(c_l^{(t-1)}(v), \{\!\{c_l^{(t-1)}(u) \mid u \in N(v)\}\!\}\right)$, where HASH is an injective mapping of the above pair to a unique value in $\Sigma$, that has not been used in previous iterations. The HASH function can be implemented by assigning consecutive integers to pairs upon first occurrence (Shervashidze et al., 2011). Let $C_l^{(t)}(G) = \{\!\{c_l^{(t)}(v) \mid v \in V(G)\}\!\}$ be the multiset of colors a graph displays in iteration $t$. The iterative coloring terminates if $|C_l^{(t-1)}(G)| = |C_l^{(t)}(G)|$, i.e., the number of colors does not change between two iterations. For testing whether two graphs $G$ and $H$ are isomorphic, the above algorithm is run in parallel on both $G$ and $H$. If $C_l^{(t)}(G) \neq C_l^{(t)}(H)$ for any $t \geq 0$, then $G$ and $H$ are not isomorphic. The label $c_l^{(t)}(v)$ in the $t$th iteration of the 1-WL test encodes the isomorphism type of the tree of height $t$

representing $v$'s $t$-hop neighborhood (D'Inverno et al., 2021; Jegelka, 2022; Schulz et al., 2022). We write $G \not\simeq_{\text{WL}^{(t)}} H$ to denote that 1-WL distinguishes $G$ and $H$ at iteration $t$.

**From Deep Sets to graph neural networks.** Deep Sets (Zaheer et al., 2017) model permutation-invariant functions, making them ideal for unordered or multi-instance data. Given a (multi)set $X = \{x_1, x_2, \ldots, x_n\}$, a function of the form $f(X) = \rho\left(\sum_{x \in X} \phi(x)\right)$ is by design invariant to the order of elements and can represent any such function for suitable choices of $\rho$ and $\phi$. These two functions, typically realized by multi-layer perceptrons (MLPs), map elements to a feature space, aggregate them and transform the result into the final output. This approach generalizes to multisets (Xu et al., 2019), reflecting both element identity and multiplicity, thus enabling permutation-invariant modeling of complex data distributions.

GNNs can be viewed as stacked neural multiset functions, where each layer aggregates and combines node features—numerical attributes assigned to nodes—reflecting the multiset nature of graph neighborhoods. This process, known as message passing (MP), applies moment functions derived from sum-pooling: $\hat{f}(\{\!\{\mathbf{x}_1, \ldots, \mathbf{x}_k\}\!\}) = \sum_{i=1}^{k} f(\mathbf{x}_i)$ where $f \colon V^d \to V^m$ is an MLP mapping elements to a vector space (Amir et al., 2024). This allows unique representations for distinct multisets, as shown in Deep Sets. Applying this principle in GNNs, an MLP in each layer's update rule distinguishes different graph structures, effectively extending Deep Sets to graphs. The update rule of this class of *moment-based* GNNs employing this strategy can be generalized as in Equation (1), where $\mathbf{h}_v^{(k)}$ is the embedding of node $v$ at layer $k$ and $\mathbf{h}_v^{(0)}$ its initial feature vector. One-hot encoded input features ensure injectivity of summation at the first layer, even without an initial MLP. For graph-level tasks, the readout function aggregates outputs across layers to compute the graph embedding $\mathbf{h}_G$, see Equation (2), where $\|$ denotes concatenation.

$$\mathbf{h}_v^{(k)} = \text{MLP}^{(k)}\left(\mathbf{h}_v^{(k-1)} + \sum_{u \in N(v)} \mathbf{h}_u^{(k-1)}\right), \quad (1)$$

$$\mathbf{h}_G = \Big\|_{k=0}^{n} \sum_{v \in V(G)} \mathbf{h}_v^{(k)}. \quad (2)$$

The update rule can be expressed in matrix form using the adjacency matrix $\mathbf{A}$ and node feature matrix $\mathbf{H}$ as $\mathbf{H}^{(k)} = \text{MLP}^{(k)}\left((\mathbf{A} + \mathbf{I}) \cdot \mathbf{H}^{(k-1)}\right)$. The Graph Isomorphism Network (GIN) (Xu et al., 2019) extends neural multiset functions to graphs, ensuring distinct graph representations and achieving expressivity equal to the 1-WL test, the upper bound of MP-based GNNs. Unlike Equation (1) (and its matrix notation), GIN's update rule includes a learnable parameter $(1 + \epsilon^{(k)})$ multiplying the ego node embeddings

$\mathbf{h}_v^{(k-1)}$, distinguishing between the feature of the node and those of its neighbors, as captured in Lemma 2.1.

**Lemma 2.1** (Sufficient condition for the 1-WL expressivity (Xu et al., 2019)). *A moment-based GNN $\Phi^{(k)}$ is as expressive as the 1-WL test after $k$ iterations, if for all layers $j \in \{1, \ldots, k\}$ the function $\mathrm{MLP}^{(j)}$ is injective on its input domain and the aggregation rule distinguishes a node's own features from the aggregated features of its neighbors.*

Although a large body of work focuses on surpassing the expressivity of GIN and the 1-WL test (Morris et al., 2023), neighborhood aggregation remains prevalent, and 1-WL distinguishes most graphs in common benchmark datasets (Morris et al., 2021; Zopf, 2022).

# 3. Expressivity and Winning Tickets

We first establish fundamental criteria that any pruning mechanism, including those generating winning (graph) lottery tickets, must satisfy to maintain prediction quality. Let $D$ be a finite sequence of tuples $(\mathbf{A}, \mathbf{X}, t)$, where graphs $G$ are represented by adjacency and feature matrices $\mathbf{A}$ and $\mathbf{X}$, with classification targets $t$. The index $l \in [0, |D|]$ refers to the $l^{\text{th}}$ tuple, denoted as $D_l = (\mathbf{A}, \mathbf{X}, t)$, with shorthand $G_l = D_l[\mathbf{A}, \mathbf{X}]$ and $t_l = D_l[t]$. Let $\Phi^{(k)}$ be an MPNN with $k$ message-passing layers as defined in Equation (1), mapping graphs $G_l$ to an embedding space $Q$. Consider $\Phi^{(k+1)}$, the same MPNN with an additional classification layer $\mathcal{C}$. Assume $\mathcal{C}$ is a perfect classifier that correctly assigns distinct embeddings from $\Phi^{(k)}$ to their respective classes, regardless of proximity in the embedding space. Specifically, for $G_a, G_b \in D$ with $t_a \neq t_b$ and $\Phi^{(k)}(G_a) \neq \Phi^{(k)}(G_b)$, it holds that: $t_a = \mathcal{C}(\Phi^{(k)}(G_a)) \neq \mathcal{C}(\Phi^{(k)}(G_b)) = t_b$. The existence of such a $\mathcal{C}$ is supported by the results of Chen et al. (2019), showing the equivalence between graph isomorphism testing and universal function approximation. For $\Phi^{(k+1)}$ to achieve the same prediction quality on a sequence of modified tuples $\widehat{D}$ (containing, e.g., $\widehat{D}_l = (\widehat{\mathbf{A}}, \mathbf{X}, t)$, where the $l^{\text{th}}$ tuple's adjacency matrix has been pruned) as on $D$, this level of distinguishability must be preserved in pruned or otherwise adapted graphs. Similarly, any modification of $\Phi^{(k)}$ to $\widehat{\Phi}^{(k)}$ (such as pruning $\Phi^{(k)}$'s trainable parameters) must maintain this distinguishability for $\widehat{\Phi}^{(k+1)}$ to perform equivalently to $\Phi^{(k+1)}$:

**Criterion 1.** *For $\Phi^{(k+1)}$ (or $\widehat{\Phi}^{(k+1)}$) to classify all graphs in $D$ (or $\widehat{D}$) correctly, $\Phi^{(k)}$ (or $\widehat{\Phi}^{(k)}$) must distinguish all pairs of non-isomorphic graphs of different classes.*

This requirement also applies to sparse initializations of the MLPs of the MP layers in $\Phi^{(k)}$, representing winning tickets in the initialization lottery, which we analyze for neural moment-based architectures. As our work focuses on such sparse initializations, the subsequent sections exclusively address this setting.

## 3.1. Critical Paths

We analyze the expressivity of neural moment-based architectures, as defined in Equation (1), by investigating the critical computational paths within the MLPs they contain. Let the *computational graph* of an $L$-layer feed-forward MLP be represented as $G = (V, E)$, where vertices $V$ denote neurons and edges $E$ their connections. Each neuron is indexed as $v^{(i_j)}$, referring to the $i^{\text{th}}$ neuron in the $j^{\text{th}}$ layer. A layer is thus a subgraph $G^{(j)} = (V^{(j)}, E^{(j)})$ with $V^{(j)} \subset V$ and $E^{(j)} \subset E$, characterized by the adjacency matrix $\mathbf{A}^{(j)} \in \{0,1\}^{|V^{(j)}| \times |V^{(j)}|}$. Defining input and output neurons as $I^{(j)} \subset V^{(j)}$ and $O^{(j)} \subset V^{(j)}$ such that $I^{(j)} \cup O^{(j)} = V^{(j)}$, the bipartite nature of $G^{(j)}$ implies that $\mathbf{A}^{(j)}$ is symmetric and can, thus, be expressed via the bi-adjacency matrix $\tilde{\mathbf{A}}^{(j)} \in \{0,1\}^{|I^{(j)}| \times |O^{(j)}|}$. The forward pass through the $j^{\text{th}}$ layer is then $\sigma(\mathbf{x}\tilde{\mathbf{A}}^{(j)} \odot \mathbf{W}^{(j)})$, where $\mathbf{W}^{(j)}$ contains edge weights $\mathcal{W}^{(j)} = \{w_{k,l} \mid (v_k, v_l) \in E^{(j)}\}$, with the Hadamard product $\odot$ and activation function $\sigma$. Let $\mathcal{W}$ represent the set of all weights across layers. A path $p$ from input neuron $v^{(i_0)}$ to output neuron $v^{(k_L)}$ consists of a sequence $(v^{(i_0)}, \ldots, v^{(k_L-1)})$, with edges $(v^{(i_j)}, v^{(k_{j+1})}) \in E$ for $j = 0, \ldots, L-1$. The total path length is $L$, and the set of all such paths is $\mathcal{P}$. Pruning in the MLP applies a binary mask $\mathbf{M}^{(j)}$ during the forward pass: $\sigma(\mathbf{x}(\mathbf{M}^{(j)} \odot \tilde{\mathbf{A}}^{(j)} \odot \mathbf{W}^{(j)}))$. Setting $\mathbf{M}_{k,l}^{(j)} = 0$ removes edge $(v_k, v_l)$ from $E^{(j)}$, yielding $\widehat{E}^{(j)} = E^{(j)} \setminus \{(v_k, v_l)\}$. Consequently, the pruned path set $\widehat{\mathcal{P}}$ becomes: $\widehat{\mathcal{P}} = \mathcal{P} \setminus \{p \in \mathcal{P} \mid (v_k, v_l) \in p\}$, removing all paths relying on the pruned edge $(v_k, v_l)$. These paths are crucial in GNN pruning, as GNNs rely on MLPs for transformations. In post-training pruning, where each MLP's weights $\mathcal{W}$ are fixed, a path is defined as *critical* if its removal degrades the GNN's classification performance. In the context of LTH, a path is critical if its removal prevents learning the task. Specifically, if no weight configuration for the remaining paths allows the GNN to perform as well as the original, the removed path is critical.

**Definition 3.1** (Critical Paths (Pre-Training)). Let $\mathcal{W}_{\Phi^{(k)}} = \bigcup_i^k \mathcal{W}_i$ be the union of weights sets of the MLPs of the $k$ MP layers of the moment-based GNN $\Phi^{(k)}$. Then, if for some quality metric $\mathcal{M}$ to be maximized and dataset $D$ it holds that no set of weights $\mathcal{W}_{\widehat{\Phi}^{(k)}}$ exists such that $\mathcal{M}(\Phi^{(k+1)}, D) \leq \mathcal{M}(\widehat{\Phi}^{(k+1)}, D)$, we say $\mathcal{P}_{\Phi^{(k)}, C} = \mathcal{P}_{\Phi^{(k)}} \setminus \mathcal{P}_{\widehat{\Phi}^{(k)}}$ is a critical path set.

The key question is which paths in GNNs are critical and how they are characterized. A path in a GNN is critical if its removal prevents the network from distinguishing two isomorphism types of different classes, regardless of the edge weights in any layer of any MLP. Any pruning mask that removes such a path (or an associated edge) cannot be a winning ticket, as it would degrade classification accu-

racy. Conversely, pruning that preserves all critical paths can, in theory—disregarding practical issues such as over-smoothing (Keriven, 2022), oversquashing (Di Giovanni et al., 2023), bottlenecks (Alon & Yahav, 2021) or vanishing gradients —constitute a winning ticket.

In the following theoretical considerations, we assume a class of injective continuously differentiable zero-fixing activations $\sigma$ with a nowhere-zero derivative for simplicity. However, our empirical results indicate that our theory generalizes to arbitrary activations for appropriately parameterized GNNs, see Section 5. Due to the permutation invariance of GNNs, all theoretical results hold up to permutation. We do not assume a perfect classifier $\mathcal{C}$ for $\Phi^{(k+1)}$, unless stated otherwise. All proofs are provided in Appendix A.

**Theorem 3.2** (Existence of maximally expressive paths). *Let D be an arbitrary finite sequence of finite, non-trivial graphs (i.e., graphs with more than zero edges and at least one non-zero feature per node). Then, for any sufficiently overparameterized moment-based GNN $\Phi^{(k)}$ with layers employing an aggregation rule that can distinguish between a node's own features and the aggregated features of its neighbors, there exist subsets of maximally expressive paths $\mathcal{P}_{\Phi^{(k)},E} \subseteq \mathcal{P}_{\Phi^{(k)}}$ for which trainable weights $\mathcal{W}_{\widehat{\Phi}^{(k)},E}$ exist such that for any $G_a, G_b \in D$ it holds that if $G_a \not\simeq_{WL^{(k)}} G_b$ then $\widehat{\Phi}^{(k)}(G_a) \neq \widehat{\Phi}^{(k)}(G_b)$.*

While the formal results of SLTH (Malach et al., 2020; da Cunha et al., 2022) make the existence of maximally expressive paths seem plausible, they are not strictly equivalent. The proven SLTH variants are limited to MLPs or CNNs applied to classification tasks. Therefore, Theorem 3.2 constitutes a novel form of LTH, to which we refer as Strong Expressive Lottery Ticket Hypothesis or SELTH.

In line with Criterion 1, it immediately follows from Theorem 3.2 that for such a $\Phi^{(k)}$, a $\Phi^{(k+1)}$ employing a perfect classifier $\mathcal{C}$ could correctly classify every $G \in D$ and consequently $\mathcal{M}(\Phi^{(k+1)}, D) \leq \mathcal{M}(\widehat{\Phi}^{(k+1)}, D)$ for $\mathcal{P}_{\widehat{\Phi}^{(k)}} = \mathcal{P}_{\Phi^{(k)},E}$. Obviously, this existence statement is devoid of practical learning considerations, which we will address later. Before delving into those aspects, we first aim to explore how this result relates to the theory of critical paths. Specifically, it further follows directly from Theorem 3.2 that in every $\mathcal{P}_{\Phi^{(k)},E}$, there exist sufficiently expressive paths $\mathcal{P}_{\Phi^{(k)},S} \subseteq \mathcal{P}_{\Phi^{(k)},E}$ for which $\mathcal{W}_{\Phi^{(k)},S}$ exist such that for $G_a, G_b \in D$ it holds that if $t_a \neq t_b$ and $G_a \not\simeq_{WL^{(k)}} G_b$ then $\Phi^{(k)}(G_a) \neq \Phi^{(k)}(G_b)$. This is an important insight, as by the definition of critical path sets, each critical path set is then a sufficiently expressive path set, and therefore, at least in a sufficiently overparameterized GNN, every critical path set is also a subset of a maximally expressive path set, for which Theorem 3.2 shows the existence.

## 3.2. Lottery Ticket Expressivity Impact on Training.

To examine the influence of expressivity on training, we consider its impact on gradient diversity (Yin et al., 2018), see Eq. (3). Originally devised for distributed training, gradient diversity has become a standard metric for assessing neural network learning in local settings as well (Kummer et al., 2023; Rajagopal et al., 2020). Specifically, low gradient diversity can slow convergence and necessitate more passes over the dataset to reach a desired accuracy. This increases training time, especially with large batch sizes, as the effective learning rate is diluted. Furthermore, models trained with low gradient diversity often converge to sharp minima, which are associated with poor generalization, particularly when using large batch sizes. Conversely, training with higher gradient diversity aligns with smoother loss surfaces and less sharp minima, improving the model's generalization, as well as convergence rates (Yin et al., 2018).

For graphs $G_i$ with label $t_i$, the weight update steps in gradient-based training are given by $\mathbf{W}_i^{(l)} = \mathbf{W}^{(l)} - \alpha \frac{\partial \mathcal{L}_i}{\partial \mathbf{W}^{(l)}}$ for a learning rate $\alpha$ and a loss function loss $\mathcal{L}$. Gradient diversity is then given as follows for gradient matrices (using the Frobenius norm as a natural generalization of the Euclidean vector norm):

$$\Delta s = \left( \sum_{i=1}^{n} \left\| \frac{\partial \mathcal{L}_i}{\partial \mathbf{W}^{(l)}} \right\|_F^2 \right) \left( \left\| \sum_{i=1}^{n} \frac{\partial \mathcal{L}_i}{\partial \mathbf{W}^{(l)}} \right\|_F^2 \right)^{-1} \quad (3)$$

For this gradient diversity $\Delta s$, we formulate the following Theorem for moment-based GNNs, which captures the influence of the geometry of the embeddings on $\Delta s$'s magnitude.

**Theorem 3.3** (Gradient Diversity and Embeddings Orthogonality). *Assume $G_1, G_2$ with labels $t_1 \neq t_2$ and let $\mathbf{H}_1^{(l-1)}, \mathbf{H}_2^{(l-1)}$ be their corresponding embeddings at layer $l-1$. Denote the rows of $\mathbf{H}_1^{(l-1)}$ as $\{\mathbf{a}_i\}$ and the rows of $\mathbf{H}_2^{(l-1)}$ as $\{\mathbf{b}_j\}$. Suppose there exist constants $0 < m \leq M < \infty$ s.t. for all $i, j$, $m \leq \|\mathbf{a}_i\|_2^2, \|\mathbf{b}_j\|_2^2 \leq M$. Then, there exists $\zeta$ s.t. $\Delta s \geq \zeta \geq 0$ with $\zeta \propto (\sum_{ij} |\cos(\beta_{ij})|)^{-1}$ with $\beta_{ij}$ being the angle between rows $\mathbf{a}_i$ and $\mathbf{b}_j$.*

The theorem easily generalizes to any number of graphs (i.e., more than two). While the theorem specifically addresses the angles between individual node embeddings of two graphs, it also carries implications for expressivity. For instance, two graphs with identical node embeddings will always have at least as many codirectional embeddings as they have nodes (provided the graphs have the same number of nodes). Conversely, if the two graphs receive distinct node embeddings, their embeddings may still be codirectional but they can also be orthogonal, contributing zero to the sum of cosines, or otherwise divergent. Therefore, a model initialized such that two graphs do not receive (partially) identical node embeddings is likely to converge faster and generalize more effectively and thus more likely to be a winning

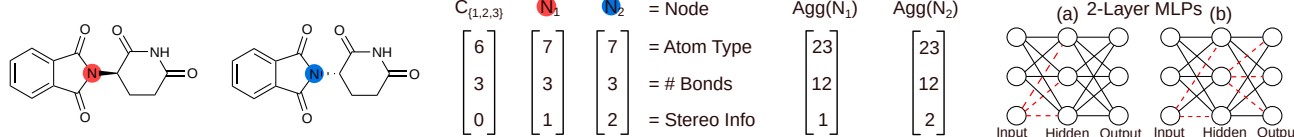

*Figure 1.* Visualization of thalidomide (also known as under its trade name Contergan) and its embryotoxic stereoisomer as an exemplary SIFDG, for which a failure to distinguish the two can have potentially life altering consequences for patients. As shown on the right-hand side of the figure, the aggregates of nodes $N_1$ and $N_2$ in the first layer differ by only a single feature. A pruning mask removing the red dashed edges from the MLPs applied to these aggregates would irrecoverably cancel this difference; hence these edges are part of critical path sets (see Definition 3.1) of a GNN trained to predict different classes for the two molecules as in, e.g., toxicity categorization tasks.

ticket in the initialization lottery, as identical embeddings are always codirectional.

In the context of Theorem 3.2, this suggests pruning a GNN $\Phi^{(k)}$ s.t. a maximally expressive path set $\mathcal{P}_{\Phi^{(k)},E}$ remains and initializing the weights $\mathcal{W}_{\widehat{\Phi}^{(k)},E}$ of this pruned $\widehat{\Phi}^{(k)}$ accordingly. Distinctness of node embeddings alone, though, does not rule out that they lie along the same line through the origin. However, depending on $\Phi^{(k)}$'s width, the following Proposition holds:

**Proposition 3.4** (Non-Colinearity under Random Pruning)**.** *For a sufficiently overparameterized moment-based GNN $\Phi^{(k)}$ with weights drawn from a continuous distribution and a finite sequence $D$ of finite input graphs, almost any random initialization combined with random pruning yields a configuration where no codirectional embeddings occur.*

Thus, the added constraint to Theorem 3.2 that for any two nodes $i, j$ with pairwise distinguishable 1-WL colors it holds that $\widehat{\Phi}^{(k)}(G_a)_i \notin \{\eta_{ij}\widehat{\Phi}^{(k)}(G_b)_j : \eta_{ij} \in \mathbb{R} \setminus \{0\}\}$ is almost always satisfied for sufficiently overparameterized $\Phi^{(k)}$.

### 3.3. Edge Cases and Boundaries of Learnability

First, we outline GNN-specific scenarios where expressivity is permanently and irrecoverably lost through pruning, regardless of the amount of training, and highlight practical cases where such a misaligned pruning could have substantial consequences. Finally, we establish bounds on the prediction quality a GNN can achieve in such cases.

**Irrecoverable cases.** Whereas for general MP layers, a reduction in expressivity is associated with a reduction in gradient diversity and thus convergence speed (see Section 3.2), an irrecoverable loss of expressivity can for certain graphs occur if the first layer is pruned incorrectly.

**Lemma 3.5** (Irrecoverable Loss of Expressivity)**.** *Consider two graphs, $G_1$ and $G_2$, with permutation-equivalent adjacency matrices but distinct feature matrices. If the pruning mask of the first layer of the MP layer's MLP—when applied in isolation—renders the graph representations indistinguishable, then no choice of weights can restore the*

*ability to distinguish between the two graphs.*

In other words, for structurally isomorphic graphs that are only distinguishable by their node feature matrices, a pruning mask that cancels the differences in the messages before the update step in the first layer will—independent of the non-zero values of the MLP's weights—render both graphs' node embeddings indistinguishable at the layer's output. As they are structurally isomorphic, as a consequence they will remain indistinguishable for the rest of the forward pass, regardless of any subsequent transformations. In the context of Theorem 3.2, Lemma 3.5 implies the removal of a path from the maximally expressive path set and, depending on the class labels $G_1$ and $G_2$, is closely related to pruning of critical paths as by Definition 3.1 and in violation of Criterion 1. We emphasize that Lemma 3.5 is considering the first layer of the first MP layer's MLP exclusively. In deeper layers, the input embeddings change during training due to changing transformations in upper layers, and what may prune away the distinction during the first iteration of training might not pose a problem in subsequent iterations. However, as Figure 1 (b) shows, the trait captured by Lemma 3.5 can extend beyond the first MLP layer, and even an extension beyond the first MP layer is plausible. Moreover, expressivity is not entirely lost if $G_1$ and $G_2$ do not have permutation-equivalent adjacency matrices, since subsequent layers can still approximate 1-WL, albeit with a different feature matrix for these graphs.

In chemical datasets, structurally similar or identical graphs (i.e., adjacency matrices identical up to permutation) can have distinct input features, leading to vastly different chemical properties (see Figure 1). Failure to distinguish such graphs after the first layer can be critical. For example, toxic stereoisomers may appear identical to their non-toxic counterparts, such as thalidomide (Lenz & Knapp, 1962; Bösl, 2014). Non-stereoisomeric cases, like Ethanol and Ethanethiol—one a beverage ingredient, the other an odorizer used in gas warning systems—illustrate this further. An improperly pruned GNN may be unable to distinguish these graphs regardless of training. We call such graphs as Structurally Isomorphic, Feature-Divergent Graphs (SIFDGs).

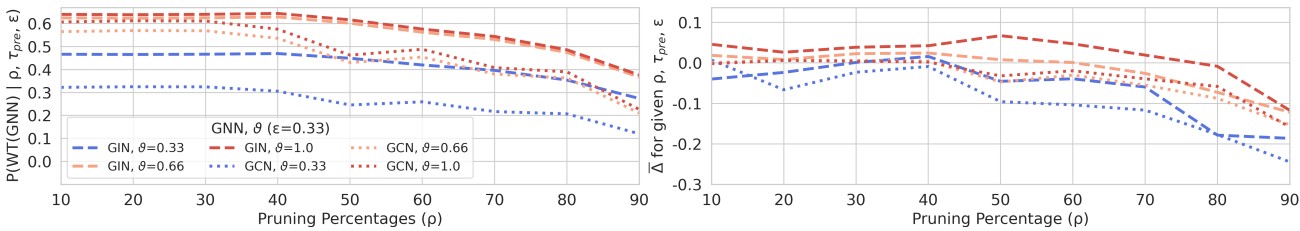

*Figure 2.* Probability that a sparse GIN or GCN is a winning ticket (WT), given pruning $\rho \in [10, 90]\%$, expressivity $\tau_{pre}$ s.t. $\vartheta - \varepsilon \leq \tau_{pre} \leq \vartheta + \varepsilon$, tolerance $\varepsilon$ (left). $\vartheta$ and $\varepsilon$ are used to group similar $\tau_{pre}$ into intervals, as observing an exact empirical value of $\tau_{pre}$ is unlikely. Mean relative test accuracy, $\overline{\Delta} = |S|^{-1} \sum_{i \in S} (A_{\text{post}}^{(i)} - A_{\text{clean}})(A_{\text{clean}})^{-1}$, where $A_{\text{clean}}$ is clean test accuracy of the dense, unpruned model, $A_{\text{post}}^{(i)}$ is post-training accuracy of pruned model $i$, and $S$ contains models with $\tau_{pre} \in [\vartheta \pm \varepsilon]$ (right). Results aggregated from training 13,500 runs over 10 datasets.

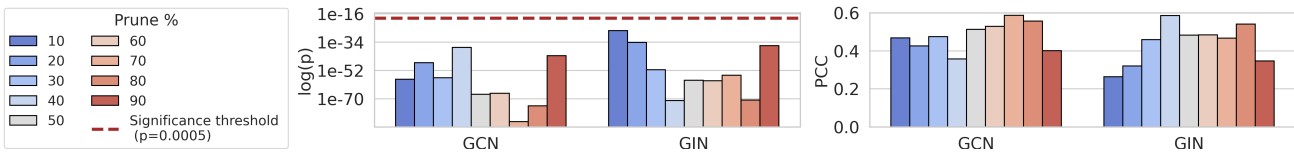

*Figure 3.* Statistical significance of Pearson correlation between untrained pruned model expressivity $\tau_{pre}$ ($\rho \in [10, 90]\%$, GIN, GCN) and post-training accuracy (left). Pearson correlation coefficients (PCC) (right). Aggregated over 13,500 runs (750 per bar, 10 datasets).

**Theoretical bounds to the quality of a lottery ticket.** Depending on the number of 1-WL distinguishable isomorphism types for which distinction in the first layer is necessary (i.e. SIFDGs) in order for the GNN to be able to learn to assign them to different classes, we can—under certain assumptions regarding class distributions—estimate how good a GNN can be if a pruning mask satisfying Lemma 3.5 is applied to the first layer.

**Lemma 3.6.** *Let $D$ be a dataset of $N$ graphs, evenly distributed across $C$ classes of $\frac{N}{C}$ graphs each. Suppose $D$ has $I$ isomorphism types, of which $U$ are indistinguishable from at least one other type by the model (i.e. $U \leq I$), covering $M$ graphs. Assuming uniform distribution, $M \approx \frac{UN}{I}$, the maximum classification accuracy is:* $1 - \left(1 - \frac{1}{C}\right) \frac{U}{I}$.

If the $U$ indistinguishable isomorphism types correspond to SIFDGs as derived from Lemma 3.5, then for any pruning mask satisfying the assumptions on data and class distribution outlined in Lemma 3.6, the above expression represents the maximal achievable accuracy for a such pruned GNN.

### 3.4. Generality and Limitations

Our theoretical results apply to moment-based GNN architectures (Section 2) in general and we thus expect the formal insights we developed—i.e., the connection between pruning, critical path removal, and loss of expressivity (e.g., Theorem 3.2, Lemma 3.5)—to apply broadly across this architectural class. This includes, for example, Graph Attention Network (Veličković et al., 2018) (GAT) or Graph Convolutional Network (GCN) (Welling & Kipf, 2016) as

well. As such, we expect our upper bounds on achievable classification accuracy under misaligned pruning (e.g., Lemma 3.6) also hold for GAT or GCN. However, refining our formal analysis to architectures beyond this most general setting (including the effects of attention or edge features that modulate aggregation) is a promising direction for future work, which might reveal additional, architecture-specific vulnerabilities not covered by our existing work.

We furthermore emphasize that Lemma 3.6 is a theoretical bound requiring knowledge of all isomorphism types of a dataset, which is impractical—though tools like nauty[1] can identify them, and popular frameworks[2] include them for certain benchmark datasets. Most datasets likely also lack the assumed uniform class distribution. The lemma is meant to conceptually illustrate how misaligned pruning limits a GNN's maximal accuracy. Refining it for more realistic settings is a potential direction for future research.

We point out that Theorem 3.3 does not directly guarantee improved convergence or generalization but instead links embedding distinctiveness to gradient diversity, a factor known to influence both. Empirically, see Section 4, since all models were trained for the same number of epochs, the consistently superior performance of sparse initializations with high expressivity suggests improved convergence and generalization, which is consistent with our theory.

Our work focuses on graph level tasks, but we expect our findings to be transferable to node level tasks as well.

---

[1] https://pallini.di.uniroma1.it/
[2] https://pytorch-geometric

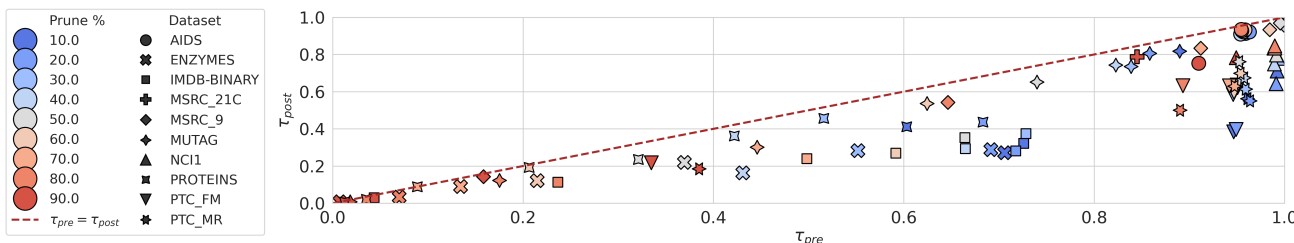

*Figure 4.* Visualization of untrained ($\tau_{pre}$) and post-training ($\tau_{post}$) expressivity of pruned models ($\rho \in [10, 90]\%$, GIN, GCN). Each marker represents the mean over the samples obtained for the models on the given dataset. The dashed line (0,0) to (1,1) indicates hypothetical pre-/post-training equivalence. Results based on 13,500 training runs in total over 10 datasets.

*Table 1.* Probability of post-training expressivity $\tau_{post} \geq \kappa$ given a pre-training expressivity $\tau_{pre} < \kappa$ for thresholds $\kappa$. Result based on 13,500 training runs in total (pruning percentages $\rho \in [10, 90]\%$, GIN, GCN) over 10 datasets.

| $P(\tau_{post} \geq \kappa \mid \tau_{pre} < \kappa)$ | 0.00 | 0.03 | 0.04 | 0.03 | 0.04 | 0.03 | 0.03 | 0.03 | 0.03 | 0.03 | 0.02 | 0.02 |
|---|---|---|---|---|---|---|---|---|---|---|---|---|
| $\kappa$ | 1.00 | 0.92 | 0.83 | 0.75 | 0.67 | 0.58 | 0.50 | 0.42 | 0.33 | 0.25 | 0.17 | 0.08 |

## 4. Experiments

We structure our experiments[3] to address the primary research question, which also drove our theoretical analysis, namely how the pre-training expressivity of a lottery ticket affects its post-training accuracy. That is, to what extent can we determine whether a sparsely initialized GNN is a winning lottery ticket based solely on its ability to distinguish non-isomorphic graphs? We investigate this research question by utilizing 10 real-world datasets from the TU-Dataset repository (Morris et al., 2020). These datasets, which are described in detail in Appendix B, are widely used in current studies. They include one-hot encoded node labels and span a variety of prevalent applications such as drug discovery (Xiong et al., 2021; Rossi et al., 2020), bioinformatics (Borgwardt et al., 2005), object recognition (Rossi & Ahmed, 2015), and social network analysis.

We assess our research question on GIN (2 hidden layers per MLP) with 2 to 4 MP layers due to its direct relation to our theory. To show our theory extends to GCN, representing the class of spectral GNNs, we include GCN in our assessment. We use ReLU activations and initialize network parameters $\mathbf{W}^{(j)}$ randomly from a uniform distribution $\mathcal{U}(-\sqrt{\frac{1}{m_j}}, \sqrt{\frac{1}{m_j}})$ with $m_j = |I^{(j)}|$, following common variance scaling initialization schemes (Glorot & Bengio, 2010; He et al., 2015). The parameterization width is matched with the input graphs' features. All models are trained for 250 epochs with a batch size of 32, a learning rate of 0.01, using the Adam optimizer. In line with LTH, only non-zero weights are updated. The experiments took approximately 8 weeks with three parallel workers to con-

clude and were conducted on a local server equipped with an NVIDIA H100 PCIe GPU (80GB VRAM), an Intel Xeon Gold 6326 CPU (500GB RAM) and a 1TB SSD.

We measure expressivity $\tau$ as graph level pre- and post-training expressivity (denoted $\tau_{pre}$ and $\tau_{post}$ in our experiments) via the percentage of non-isomorphic graphs of a dataset for which the GNN's final MP layer (at depth $n$) outputs node embeddings that are distinguishable for FLOAT32 ($\epsilon_{mach} = 1.19 \times 10^{-7}$). Specifically, we retain one representative per isomorphism type, as isomorphic graphs yield identical embeddings by GNN permutation invariance. Let $\{G_1, \ldots, G_m\}$ be $m$ such representatives with embeddings $\{\mathbf{h}_{G_1}, \ldots, \mathbf{h}_{G_m}\}$, $\mathbf{h}_{G_i} = \sum_{v \in V(G_i)} \mathbf{h}_v^{(n)}$. We mark each pair $(\mathbf{h}_{G_i}, \mathbf{h}_{G_j})$ as indistinguishable if $\mathbf{h}_{G_i} - \mathbf{h}_{G_j} = \mathbf{0}$, which generally implies differing node-level embeddings: while distinct node embeddings can theoretically sum to the same vector, such collisions are extremely rare and occur only under measure-zero configurations. The expressivity $\tau$ is the fraction that remains distinguishable. Alternative methods (e.g., exhaustive comparisons or training accuracy) are more costly or reflect different notions of expressivity. We chose our approach for its scalability and direct assessment of whether a graph is distinguishable from the rest of the dataset. A sparsely initialized model is a winning ticket if its test accuracy degrades by less than 0.05 relative to its dense counterpart.

## 5. Results

Our experiments empirically confirm both our theoretical predictions made in Section 3. Specifically, our experiments show that sparse yet highly expressive (even if their weights are untrained), trainable subnetworks exist (Theorem 3.2).

---

[3]The code for reproducing our results is available at GitHub:
https://github.com/lorenz0890/wl2025lottery

Moreover, they show that for a given percentage of random weights pruning, the expressivity of the pruned network before training is at least one of the driving factors of its post-training accuracy.

The probability that a GNN is a winning ticket given a certain pruning percentage and pre-training expressivity is high for high expressivity and low otherwise for all but the highest pruning ratios (Figure 2, left). To compute this probability, for each pruning ratio $\rho$, we set a target $\vartheta$ and collect runs with $\tau_{pre} \in [\vartheta \pm \varepsilon]$. A run is labeled a "winning ticket" if its accuracy decreases by less than $0.05$ relative to the unpruned model. The probability is the fraction of these runs, aggregated (with subset-size normalization) to visualize winning ticket probabilities across pruning levels and thresholds. Given that we trained all our models for the same number of epochs, our results are consistent with the hypothesis put forward in Section 3.2 that an increased expressivity in the initialization potentially improves model convergences and generalization, as is indicated by Theorem 3.3. Up to 80% pruning, highly expressive sparse GINs outperformed dense unpruned models in post-training prediction, while less expressive models failed to match dense model quality (Figure 2, right).

Moreover, we find that a GNN, when initialized with a certain sparsity and non-zero weights trained in isolation, is highly unlikely to gain expressivity (Table 1). That is, if a GNN initialized, pruned and trained as described could reliably transition from low $\tau_{pre}$ to high $\tau_{post}$, then for some $\kappa$, this transition would occur with a probability exceeding a low single-digit percentage. Although we do not explicitly set a threshold defining "high probability", Figure 4 illustrates the trend outlined in Table 1. Consistent with Table 1, Figure 4 shows that sparse GNNs rarely transition from low to high expressivity during training. Instead, the converse is the typical case, as all data points fall below the dashed $\tau_{post} = \tau_{pre}$ line, indicating that $\tau_{post}$ is generally lower than $\tau_{pre}$, whereby low pruning percentages apparently exaggerate the effect. This aligns with Lemma 3.5 (focused on the input layer) and highlights the broader trend suggested in Section 3.3: despite the dataset-dependent differences apparent in Figure 4, recovery from a relatively low expressivity sparse initialization is rare during training if only non-zero weights are updated, even if theoretically possible. As pre-training expressivity was low even at conservative pruning rates on some datasets, we interpret this, following Proposition 3.4, as an indication that the underlying GNNs were insufficiently parameterized for the pruning rate.

Finally, Figure 3 reveals an intriguing pattern: the statistical significance of the correlation between pre-training expressivity and post-training prediction quality (left) is lowest for mid to slightly above mid-pruning ratios but higher for very low and high pruning ratios (though still significant). Conversely, the Pearson correlation coefficients (right) show an inverse trend. We conjecture this reflects the greater likelihood of irrecoverable damage (compare Section 3.3) at high pruning ratios, while achieving good post-training prediction quality is generally more probable at low pruning ratios and less related to expressivity, leading to increased p-values and decreased correlation coefficients in both cases.

**Implications for practitioners and future research.** As demonstrated in this work, the expressivity of sparsely initialized GNNs influences convergence speed and generalization quality. Irrecoverable pruning scenarios, where degraded expressivity cannot be restored through training, underscore the need for careful pruning design. Preserving critical paths is essential to avoid catastrophic errors, such as misclassifying toxic and non-toxic stereoisomers.

Future work should seek to enhance convergence, generalization, and optimization by developing sparse yet expressive initializations. For moment-based GNNs, injective aggregate and combine functions over dataset-defined domains are sufficient for maximal expressivity (i.e., 1-WL -equivalence for GIN). A simple pre-training sparsification strategy could proceed layer-wise: for a fixed pruning ratio, sample $k$ configurations and retain the one preserving input–output injectivity across all graphs. This is repeated per layer, increasing sparsity until no injective configuration is found within $k$ trials. The result is a maximally expressive, sparsified network. Adapting our results on winning tickets to settings such as those explored by Wałęga & Rawson (2025), Bause et al. (2024) or Kummer et al. (2024) is another promising direction. Pruning strategies that prioritize expressivity could improve LTH's practicality for GNNs, ensuring reliability and performance in critical domains.

# 6. Conclusion

In this work, we bridge the gap between GNN expressivity and LTH. We offer theoretical insights and empirical validation that trainable sparse initializations with comparable expressivity to dense models exist. Moreover, we show that increased expressivity in the initialization potentially accelerates model convergence and improves generalization. Our findings highlight the risk of pruning strategies that fail to retain critical computational paths, in certain cases theoretically causing irreversible degradation in performance. Recovering expressivity loss induced by sparsity through training is generally unlikely, underscoring the need for sparsification approaches that safeguard key expressive capabilities. Our work also has practical implications: preserving critical paths during pruning is essential to prevent catastrophic errors, such as misclassifying toxic and non-toxic stereoisomers, particularly in critical applications like drug discovery or molecular property prediction.

## Acknowledgements

This work was supported in part by the Vienna Science and Technology Fund (WWTF) and the City of Vienna through grants [10.47379/VRG19009] and [10.47379/ICT22059].

## Impact Statement

Our work highlights the critical role of preserving expressivity in sparsely initialized and trained GNNs, offering insights into how pruning can lead to irrecoverable expressivity loss. By demonstrating the risks associated with degraded expressivity, we show the importance of careful pruning design to ensure reliable predictions, such as distinguishing toxic from non-toxic compounds in drug discovery. Moreover, via linking GNN expressivity with generalization and convergence, we show that degraded expressivity can hinder both training dynamics and model reliability. The broader societal benefit of our work lies in its potential to inform the development of high-performance and trustworthy GNNs in resource-constrained settings, contributing to equitable access to advanced machine learning technologies and promoting their safe use in impactful domains.

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

# A. Proofs

## A.1. Proof of Theorem 3.2

*Proof.* The proof consists of two parts. In the first part, we show the existence of subsets paths for which weights exist such that any 1-WL distinguishable pair of graphs of a bounded dataset can be distinguished. In the second part, we show that the weights of these paths are trainable, i.e., that they can receive gradient updates.

As established in the literature (Lemma 2.1), a neural moment-based GNN is maximally expressive if all of its MLPs are injective on their respective input domains; that is, if distinct inputs representing neighborhood aggregates are mapped to distinct outputs. Therefore, it is necessary to demonstrate the existence of sparse subnetworks capable of modeling such injective functions. Noting that an MLP is injective if all of its layers are injective, we begin by proving the following Lemma:

**Lemma A.1** (Layer Injectivity under Random Pruning). *Let $\mathbf{W} \in \mathbb{R}^{m \times n}$ with $1 < n \leq m$ be a linear transformation matrix where each entry is independently drawn from a continuous and bounded distribution over an interval $[a, b]$ with $a < b$. Let $\sigma : \mathbb{R} \to \mathbb{R}$ an injective elementwise activation function. Consider a finite inpute set $X \subset \mathbb{R}^n$ with cardinality $N$ where each element has continous entries bounded by $[c, d]$. Suppose we randomly prune $\mathbf{W}$ by independently setting each weight to zero with probability $\rho$ (sparsity ratio), resulting in a sparse weight matrix $\mathbf{W}'$. Then, the function $f = \sigma \circ \mathbf{W}' : X \to \mathbb{R}^m$ is injective with probability $\gamma \geq 1 - \binom{N}{2} \rho^{km}$ where $k \geq 1$ is the minimum number of non-zero components in $\mathbf{x}_u - \mathbf{x}_v$ for all distinct pairs $\mathbf{x}_u, \mathbf{x}_v \in X$*

*Proof.* The function $f$ fails to be injective if there exist distinct inputs $\mathbf{x}_u, \mathbf{x}_v \in X$ s.t. $f(\mathbf{x}_u) = f(\mathbf{x}_v)$. Since $\sigma$ is injective, this implies that $\mathbf{W}'\mathbf{x}_u = \mathbf{W}'\mathbf{x}_v$ Let $\delta = \mathbf{x}_u - \mathbf{x}_v \neq \mathbf{0}$. Then, the condition $\mathbf{W}'\mathbf{x}_u = \mathbf{W}'\mathbf{x}_v$ becomes $\mathbf{W}'\delta = \mathbf{0}$.

For a fixed $\delta$, we compute the probability that $\mathbf{W}'\delta = \mathbf{0}$. Let $k$ be the number of non-zero components in $\delta$, i.e., $k = \|\delta\|_0 \geq 1$. Each weight $w'_{rj}$ in $\mathbf{W}'$ is then given by $w'_{rj} = w_{rj} z_{rj}$ where $w_{rj}$ is the original weights and $z_{rj}$ is an independent Bernoulli random variable:

$$z_{rj} = \begin{cases} 0, & \text{with probability } \rho, \\ 1, & \text{with probability } 1 - \rho. \end{cases}$$

For each row $r$ of $\mathbf{W}'$, the $r$-th component of $\mathbf{W}'\delta$ is:

$$(\mathbf{W}'\delta)_r = \sum_{j=1}^n w'_{rj}\delta_j = \sum_{j=1}^n w_{rj} z_{rj}\delta_j.$$

**If all** $z_{rj}\delta_j = 0$**:** This occurs if either $z_{rj} = 0$ (due to pruning) or $\delta_j = 0$.

- The probability that $z_{rj} = 0$ when $\delta_j \neq 0$ is $\rho$.

- Since $z_{rj}$ are independent and there are $k$ non-zero $\delta_j$, the probability that all $z_{rj} = 0$ for those $j$ is $\rho^k$.

**If any** $z_{rj}\delta_j \neq 0$**:** Since $w_{rj}$ are continuous random variables over $[a, b]$ and $\delta_j \neq 0$, the product $w_{rj}\delta_j$ is a continuous random variable. Therefore, the sum $(\mathbf{W}'\delta)_r$ is also a continuous random variable, and the probability that it equals zero is zero. That is, the set of weights elements of each row vector $\mathbf{w}_r$ satisfying $\mathbf{w}_r^T\delta = 0$ forms a hyperplane in $\mathbb{R}^n$. Since elements $w_{rj}$ are drawn independently from a continuous distribution, the probability that they fall on that hyperplane is practically 0, which is equivalent to every proper hyperplane having a zero Lebesgue measure in its ambient space.

Thus, the probability $(\mathbf{W}'\delta)_r = 0$ is

$$P\left((\mathbf{W}'\delta)_r = 0\right) = \rho^k.$$

Therefore, as the rows of $\mathbf{W}'$ are independent because the $w'_{rj}$ are independent across $r$, the probability that $\mathbf{W}'\delta = \mathbf{0}$ is

$$P\left(\mathbf{W}'\delta = \mathbf{0}\right) = \left(\rho^k\right)^m = \rho^{km}.$$

There are $\binom{N}{2}$ distinct pairs $(\mathbf{x}_u, \mathbf{x}_v)$ in $X$. By Boole's inequality, the probability that there exists at least one pair $(\mathbf{x}_u, \mathbf{x}_v)$ such that $\mathbf{W}'\delta = \mathbf{0}$ is now at most:

$$P\left(\exists \delta \neq \mathbf{0} : \mathbf{W}'\delta = \mathbf{0}\right) \leq \binom{N}{2} \rho^{km}.$$

Therefore, the probability that $f$ is injective over $X$ is at least:

$$\gamma = 1 - \binom{N}{2} \rho^{km}.$$

$\square$

Since $k \geq 1$ and $\rho \in (0, 1)$, $\rho^{km}$ decreases exponentially with $m$, and for sufficiently large $m$, $\gamma$ will always be $> 0$. That is, for $\rho > 0, \gamma > 0$ and for fixed $N, k$, we choose $m$ such that $m \geq \log_\rho\left((1 - \gamma)\binom{N}{2}^{-1}\right) k^{-1}$.

Then, it immediately follows that for an MLP with $L$ layers, the probability of a random pruning leaving it injective is $\gamma_L \geq (1 - \binom{N}{2}\rho^{km_{min}})^L$ with $m_{min}$ being the smallest number of neurons of any of the $L$ layers. Note that treating the layers independently is conservative, as non-injectivity in a previous layer would decrease $N$ (size of distinct elements in input) for the subsequent layer and thus increase the probability of it being injective w.r.t to these inputs.

Likewise, we can model a GNN on a bounded dataset $D$ of finite graphs. If there are at most $N$ nodes in any graph in $D$ and we have $M$ MP layers, the total probability for the pruned GNN having injective MLPs is then $\gamma_{GNN} \geq (1 - \binom{|D|N}{2}\rho^{km_{min}})^{LM}$, which corresponds to the probability that a non-zero pruning rate (corresponding to pruning masks corresponding to edge- and therefore computational path-deletions) leaves a subset of paths intact in the GNN for which weights exist such that the GNN is maximally expressive (in the sense of Lemma 2.1) on the given dataset $D$, i.e., if $G_a \not\simeq_{WL^{(k)}} G_b, G_a, G_b \in D$ then $\widehat{\Phi}^{(k)}(G_a) \neq \widehat{\Phi}^{(k)}(G_b)$, given a sufficient number of neurons $m$ for each of the MLP layers.

Based on the above shown existence of $\mathcal{P}_{\widehat{\Phi}^{(k)}, E} \subseteq \mathcal{P}_{\Phi^{(k)}}$ and $\mathcal{W}_{\widehat{\Phi}^{(k)}, E}$ such that any $G_a, G_b \in D, G_a \not\simeq_{WL^{(k)}} G_b$ can be distinguished via $\widehat{\Phi}^{(k)}(G_a) \neq \widehat{\Phi}^{(k)}(G_b)$, we now show that these weights can receive weights updates. For simplicity and to focus on the most relevant interactions between trainable parameters and other components of MP, the proof only considers MP layers with a single layer MLP (i.e., not hidden layers), but it can easily be generalized to an arbitrary number of layers per MLP.

**Lemma A.2** (Trainability). *Let $\widehat{\Phi}^{(k)}$ be a GNN pruned to an $\mathcal{P}_{\widehat{\Phi}^{(k)}, E}$ initialized to appropriate $\mathcal{W}_{\widehat{\Phi}^{(k)}, E}$ for a dataset $D$, where $D$ consists of bounded, non-trivial graphs (e.g., graphs with more than zero edges and at least one non-zero feature per node) Then, for any loss $\mathcal{L}(\widehat{\Phi}^{(k+1)}(G), t) \neq 0$, at least those weights relevant to the distinction of $G$ from other $H \in D$ are capable of receiving updates.*

*Proof.* The general equations for backpropagation through a GNN with $\mathbf{Z}^{(l)} = \mathbf{A}\mathbf{H}^{(l-1)}\mathbf{W}^{(l)}$ and $\mathbf{H}^{(l)} = \sigma(\mathbf{Z}^{(l)})$ of the form Equation (1), where $\frac{\partial \mathbf{Z}^{(l)}}{\partial \mathbf{H}^{(l)}} = \mathbf{W}^{(l)T}$, $\frac{\partial \mathcal{L}}{\partial \mathbf{H}^{(l)}} = \mathbf{A}^T \frac{\partial \mathcal{L}}{\partial \mathbf{Z}^{(l+1)}} \mathbf{W}^{(l+1)T}$, and $\frac{\partial \mathcal{L}}{\partial \mathbf{W}^{(l)}}$ are derived via chain rule as

$$
\begin{aligned}
\frac{\partial \mathcal{L}}{\partial \mathbf{W}^{(l)}} &= \frac{\partial \mathcal{L}}{\partial \mathbf{H}^{(l)}} \frac{\partial \mathbf{H}^{(l)}}{\partial \mathbf{W}^{(l)}} \\
&= \frac{\partial \mathcal{L}}{\partial \mathbf{H}^{(l)}} \frac{\partial \mathbf{H}^{(l)}}{\partial \mathbf{Z}^{(l)}} \frac{\partial \mathbf{Z}^{(l)}}{\partial \mathbf{W}^{(l)}} \\
&= \mathbf{H}^{(l-1)T} \mathbf{A}^T \frac{\partial \mathcal{L}}{\partial \mathbf{Z}^{(l)}} \\
&= \mathbf{H}^{(l-1)T} \mathbf{A}^T \left(\sigma'(\mathbf{Z}^{(l)}) \odot \frac{\partial \mathcal{L}}{\partial \mathbf{H}^{(l)}}\right) \\
&= \mathbf{H}^{(l-1)T} \mathbf{A}^T \left(\sigma'(\mathbf{Z}^{(l)}) \odot \mathbf{A}^T \frac{\partial \mathcal{L}}{\partial \mathbf{Z}^{(l+1)}} \mathbf{W}^{(l+1)T}\right)
\end{aligned}
\tag{4}
$$

which, written as elementwise summation, yields

$$\frac{\partial \mathcal{L}}{\partial W_{ij}^{(l)}} = \sum_{l=1}^{d_l} \sum_{r=1}^{n} \sum_{p=1}^{n} \sum_{q=1}^{n} H_{qi}^{(l-1)} A_{pq} A_{rp} \sigma'(Z_{pj}^{(l)}) \frac{\partial \mathcal{L}}{\partial Z_{rl}^{(l+1)}} W_{jl}^{(l+1)} \tag{5}$$

To show back-propagation works for these paths, we show by induction the existence of non-zero gradients at every layer.

**Base Case.** By assumption $\widehat{\Phi}^{(k)}$ is pruned to an expressive path set $\mathcal{P}_{\widehat{\Phi}^{(k)},E}$ initialized to appropriate $\mathcal{W}_{\widehat{\Phi}^{(k)},E}$, allowing the distinction of WL distinguishable graphs. That means for any $G$ in $D$, there must be at least one node embedding at the $k^{\text{th}}$ layers output of $\widehat{\Phi}^{(k)}$ that distinguishes it from those $H \in D$ that are $H \not\sim_{WL^{(k)}} G$, as otherwise either $\mathcal{P}_{\widehat{\Phi}^{(k)},E}$ would not be a maximally expressive path set or $\mathcal{W}_{\widehat{\Phi}^{(k)},E}$ would not be properly initialized as per Theorem 3.2. That is, at least one of the two graphs $G$ or $H$ must have at least one non-zero node embeddings and, for the sake of the argument, we let this graph be $G$. Let's further say $p$ is one of those nodes of $G$ where embeddings are distinguishing $G$ from $H$.Then, this node $p$ must have at least one non-zero embedding $i$ at layer $k$, i.e., $\sigma(Z_{pi}^{(k)}) \neq 0$,

Now, we start at layer $l = k$, right before the classifier layer. Assume $\frac{\partial \mathcal{L}}{\partial Z_{rt}^{(k+1)}} W_{it}^{(k+1)} \neq 0$, i.e. loss back propagated for the $t^{\text{th}}$ feature of the $r^{\text{th}}$ node from the classifier $\mathcal{C}$ used at layer $k + 1$ and the $t^{\text{th}}$ weight of the $i^{\text{th}}$ neuron of the classifier is also non-zero (which it is since we assume a dense classifier with random continuous weights).

Then we know by assumption $\sigma(Z_{pi}^{(k)}) \neq 0$ that for some node $p$ connected to $r$ (or at least $r = p$) at least one feature $i$ of node $p$ must have had a non-zero contribution to $\sigma(Z_{pi}^{(k)}) \neq 0$ as $\sigma$ is injective and zero-fixing (i.e. only $\sigma(0) = 0$ ) and consequently, as $\sigma$ has nowhere-zero derivative, at least for these indices,

$$\frac{\partial \mathcal{L}}{\partial Z_{pi}^{(k)}} = \sigma'(Z_{pi}^{(k)}) \sum_{t=1}^{d_k} \sum_{r=1}^{n} A_{rp} \frac{\partial \mathcal{L}}{\partial Z_{rt}^{(k+1)}} W_{it}^{(k+1)} \neq 0. \tag{6}$$

From the assumption $\sigma(Z_{pi}^{(k)}) \neq 0$ and $\sigma$ being zero-fixing it further follows that

$$Z_{pi}^{(k)} = \sum_{q}^{n} \sum_{j}^{d_{k-1}} A_{pq} H_{qj}^{(k-1)} W_{ji}^{(k)} \neq 0 \tag{7}$$

which implies that for some node $q$ connected to $r$ (or $r = q$) the $j^{\text{th}}$ feature had a non-zero embedding at layer $k - 1$, i.e., $H_{qj}^{(k-1)} \neq 0$, and moreover, $W_{ji}^{(k)} \neq 0$ (which in turn implies inclusion of the edge $ji$ of the computational graph in the expressive path set $\mathcal{P}_{\widehat{\Phi}^{(k)},E}$).

Thus, for the gradient update step,

$$\frac{\partial \mathcal{L}}{\partial W_{ji}^{(k)}} = \sum_{t=1}^{d_l} \sum_{r=1}^{n} \sum_{p=1}^{n} \sum_{q=1}^{n} H_{qj}^{(k-1)} A_{pq} A_{rp} \sigma'(Z_{pi}^{(k)}) \frac{\partial \mathcal{L}}{\partial Z_{rt}^{(k+1)}} W_{it}^{(k+1)} \tag{8}$$

we find that indices exist which for which the non-zero weight $W_{ji}^{(k)}$ as part of the expressive path set receives a nonzero gradient and the loss gradient $\frac{\partial \mathcal{L}}{\partial Z_{pi}^{(k)}}$ that can be back propagated to the next layer.

**Assumption.** For some $su$ and $iu$ $\frac{\partial \mathcal{L}}{\partial Z_{su}^{(l+1)}} W_{iu}^{(l+1)} \neq 0$.

**Step.** In the induction step, we show that the induction assumption implies that the error is back propagated through layer $l - 1$ and the weights at layer $l - 1$ receive updates via the recursive definition of backpropagation:

$$\frac{\partial \mathcal{L}}{\partial W_{ji}^{(l-1)}} = \sum_{t=1}^{d_{l-1}} \sum_{r=1}^{n} \sum_{p=1}^{n} \sum_{q=1}^{n} \sum_{u=1}^{d_l} \sum_{s=1}^{n} H_{qj}^{(l-2)} A_{pq} A_{rp} A_{sr} \sigma'(Z_{pi}^{(l-1)}) \sigma'(Z_{rt}^{(l)}) \frac{\partial \mathcal{L}}{\partial Z_{su}^{(l+1)}} W_{iu}^{(l+1)} W_{it}^{(l)} \tag{9}$$

By the induction assumption, for some $su$ and $iu$, $\frac{\partial \mathcal{L}}{\partial Z_{su}^{(l+1)}} W_{iu}^{(l+1)} \neq 0$. As we also assume that $\sigma(Z_{pi}^{(k)}) \neq 0$ for some $pi$, also $\sigma(Z_{rt}^{(l)}) \neq 0$ for some $rt$ as otherwise, no non-zero forward propagation could have happened. By the same argument, for some $pi$, $\sigma(Z_{pi}^{(l-1)}) \neq 0$, implying $\frac{\partial \mathcal{L}}{\partial Z_{pi}^{(l-1)}} \neq 0$ for some node $p$. Thus, assuming $W_{ji}^{(l-1)} \neq 0$, as the edge is part the expressive path set, it follows that $H_{qj}^{(l-2)} \neq 0$ for some $qj$. Then, if nodes $pq$, $rp$ and $sr$ are connected or have self loops, at least for those node indices, $\frac{\partial \mathcal{L}}{\partial W_{ji}^{(l-1)}} \neq 0$. $\qquad\square$

To summarize, we have shown that it follows from Lemma A.1 that for any arbitrary finite sequence of finite graphs $D$ (e.g., graphs with more than zero edges and at least one non-zero feature per node). and any sufficiently overparameterized moment-based GNN $\Phi^{(k)}$ with layers employing an aggregation rule that can distinguish between a node's own features and the aggregated features of its neighbors, there exist subsets of maximally expressive paths $\mathcal{P}_{\Phi^{(k)},E} \subseteq \mathcal{P}_{\Phi^{(k)}}$ for which weights $\mathcal{W}_{\widehat{\Phi}^{(k)},E}$ exist such that for any $G_a, G_b \in D$ it holds that if $G_a \not\simeq_{WL^{(k)}} G_b$ then $\widehat{\Phi}^{(k)}(G_a) \neq \widehat{\Phi}^{(k)}(G_b)$ and, from Lemma A.2, that these $\mathcal{W}_{\widehat{\Phi}^{(k)},E}$ for $\mathcal{P}_{\Phi^{(k)},E} \subseteq \mathcal{P}_{\Phi^{(k)}}$ are trainable. $\qquad\square$

### A.2. Proof of Theorem 3.3

*Proof.* Let $G_i$ with labels $t_i$ be graphs and $\mathcal{L}_i$ be the loss w.r.t to the input graph $G_i$ with label $t_i$. Then, gradient diversity as given by (3) via Frobenius inner product expansion, can be rewritten as

$$\Delta s = \Big( \sum_{i=1}^{n} \|\frac{\partial \mathcal{L}_i}{\partial \mathbf{W}^{(l)}}\|_F^2 \Big) \Big( \sum_{i=1}^{n} \|\frac{\partial \mathcal{L}_i}{\partial \mathbf{W}^{(l)}}\|_F^2 + \sum_{i \neq j} \Big\langle \frac{\partial \mathcal{L}_i}{\partial \mathbf{W}^{(l)}}, \frac{\partial \mathcal{L}_j}{\partial \mathbf{W}^{(l)}} \Big\rangle_F \Big)^{-1}. \tag{10}$$

Obviously, the Frobenius inner product (a generalization of the dot product to matrices) between gradients of different data points dictates how large or small the gradients' diversity is: if all gradients are orthogonal , it is maximal, whereas if they are codirectional, it is minimal.

Let's now consider two graphs specific $G_1, G_2$ with labels $t_1 \neq t2$ and analyze this inner product product. Then, with

$$\frac{\partial \mathcal{L}_i}{\partial \mathbf{W}^{(l)}} = \mathbf{H}_i^{(l-1)T} \mathbf{A}_i^T \frac{\partial \mathcal{L}}{\partial \mathbf{Z}_i^{(l)}} \tag{11}$$

from Equation (4) we obtain for the inner product

$$tr\Big( \Big( \mathbf{H}_1^{(l-1)T} \mathbf{A}_1^T \frac{\partial \mathcal{L}_1}{\partial \mathbf{Z}_1^{(l)}} \Big)^T \mathbf{H}_2^{(l-1)T} \mathbf{A}_2^T \frac{\partial \mathcal{L}_2}{\partial \mathbf{Z}_2^{(l)}} \Big) = \Big\langle \mathbf{H}_1^{(l-1)T} \mathbf{A}_1^T \frac{\partial \mathcal{L}_1}{\partial \mathbf{Z}_1^{(l)}}, \mathbf{H}_2^{(l-1)T} \mathbf{A}_2^T \frac{\partial \mathcal{L}_2}{\partial \mathbf{Z}_2^{(l)}} \Big\rangle_F \tag{12}$$

which (via transposition and cyclicity of the trace of matrix products) is equal to

$$tr\Big( \mathbf{H}_1^{(l-1)} \mathbf{H}_2^{(l-1)T} \mathbf{A}_2^T \frac{\partial \mathcal{L}_2}{\partial \mathbf{Z}_2^{(l)}} \frac{\partial \mathcal{L}_1}{\partial \mathbf{Z}_1^{(l)}} \mathbf{A}_1 \Big) = tr\Big( \Big( \mathbf{H}_1^{(l-1)T} \mathbf{A}_1^T \frac{\partial \mathcal{L}_1}{\partial \mathbf{Z}_1^{(l)}} \Big)^T \mathbf{H}_2^{(l-1)T} \mathbf{A}_2^T \frac{\partial \mathcal{L}_2}{\partial \mathbf{Z}_2^{(l)}} \Big). \tag{13}$$

For this left hand side, without assumptions of definiteness of the matrices involved, an upper bound of it's magnitude (i.e. absolute value) is provided by

$$|tr\Big( \mathbf{H}_1^{(l-1)} \mathbf{H}_2^{(l-1)T} \mathbf{A}_2^T \frac{\partial \mathcal{L}_2}{\partial \mathbf{Z}_2^{(l)}} \frac{\partial \mathcal{L}_1}{\partial \mathbf{Z}_1^{(l)}} \mathbf{A}_1 \Big)| \leq \|\mathbf{H}_1^{(l-1)} \mathbf{H}_2^{(l-1)T}\|_F \cdot \|\mathbf{A}_2^T \frac{\partial \mathcal{L}_2}{\partial \mathbf{Z}_2^{(l)}} \frac{\partial \mathcal{L}_1}{\partial \mathbf{Z}_1^{(l)}} \mathbf{A}_1\|_F. \tag{14}$$

The first part of this product itself can be bounded by exploiting the relation between the Frobenius norm and the inner product

$$\|\mathbf{H}_1^{(l-1)} \mathbf{H}_2^{(l-1)T}\|_F \leq \sqrt{(\max_{i,j} \|\mathbf{a}_i\|_2^2 \|\mathbf{b}_j\|_2^2) \sum_{i,j} \cos^2(\beta_{ij})} \tag{15}$$

with $i, j$ denoting rows of $\mathbf{H}_1^{(l-1)}, \mathbf{H}_2^{(l-1)}$ and $\mathbf{a}_i, \mathbf{b}_j$ the corresponding row vectors and $\beta_{ij}$ the angle between each pair $\mathbf{a}_i, \mathbf{b}_j$ and $\sum_{i,j}$ the double sum over the common dimension of the two matrices. Thus, it follows that if all node embeddings are orthogonal, gradient diversity is maximal as their enclosing angles' cosines equal 0.

Furthermore, if there exist $m, M$ s.t. for all $i, j$ it holds that $m \leq ||\mathbf{a}_i||_2^2, ||\mathbf{b}_j||_2^2 \leq M$, then $\max_{i,j} ||\mathbf{a}_i||_2^2 ||\mathbf{b}_j||_2^2 \leq M^2$ and thus $||\mathbf{H}_1^{(l-1)} \mathbf{H}_2^{(l-1)T}||_F \leq M \sqrt{\sum_{i,j} \cos^2(\beta_{ij})}$, which is proportional up to a constant factor $M$ to the sum of the cosine similarity of the node embeddings.

Consequentially, we can formulate for the two graphs $G_1, G_2$

$$\zeta_{\pm} = \Big(\sum_{i=1}^{2}||\frac{\partial \mathcal{L}_i}{\partial \mathbf{W}^{(l)}}||_F^2\Big)\Big(\sum_{i=1}^{2}||\frac{\partial \mathcal{L}_i}{\partial \mathbf{W}^{(l)}}||_F^2 \pm M \sqrt{\sum_{i,j} \cos^2(\beta_{ij})} \cdot ||\mathbf{A}_2^T \frac{\partial \mathcal{L}_2}{\partial \mathbf{Z}_2^{(l)}} \frac{\partial \mathcal{L}_1}{\partial \mathbf{Z}_1^{(l)}} \mathbf{A}_1||_F\Big)^{-1}. \tag{16}$$

for which it holds that for any $\sum_{i=1}^{2}||\frac{\partial \mathcal{L}_i}{\partial \mathbf{W}^{(l)}}||_F^2 \neq 0$ either $\Delta s \in [\zeta_+, \infty)$ if $0 < \sum_{i=1}^{2}||\frac{\partial \mathcal{L}_i}{\partial \mathbf{W}^{(l)}}||_F^2 < M \sqrt{\sum_{i,j} \cos^2(\beta_{ij})} \cdot ||\mathbf{A}_2^T \frac{\partial \mathcal{L}_2}{\partial \mathbf{Z}_2^{(l)}} \frac{\partial \mathcal{L}_1}{\partial \mathbf{Z}_1^{(l)}} \mathbf{A}_1||_F$ or $\Delta s \in [\zeta_+, \zeta_-]$ if $\sum_{i=1}^{2}||\frac{\partial \mathcal{L}_i}{\partial \mathbf{W}^{(l)}}||_F^2 \geq M \sqrt{\sum_{i,j} \cos^2(\beta_{ij})} \cdot ||\mathbf{A}_2^T \frac{\partial \mathcal{L}_2}{\partial \mathbf{Z}_2^{(l)}} \frac{\partial \mathcal{L}_1}{\partial \mathbf{Z}_1^{(l)}} \mathbf{A}_1||_F$.

Thus, for any $\Delta s$, we obtain the proposed lower bound by choosing $\zeta = \zeta_+$ for which it directly follows from equation (16) that $\zeta \propto (\sum_{ij} |cos(\beta_{ij})|)^{-1}$. $\qquad \square$

### A.3. Proof of Proposition 3.4

*Proof.* Let the function

$$f = \sigma \circ \mathbf{W}' : X \to \mathbb{R}^m,$$

where $\mathbf{W} \in \mathbb{R}^{m \times n}$, $1 < n \leq m$, with entries independently drawn from a continuous and bounded distribution over an interval $[a, b]$ with $a < b$. We use an injective elementwise activation $\sigma : \mathbb{R} \to \mathbb{R}$. Suppose we randomly prune $\mathbf{W}$ by independently setting each weight to zero with probability $\rho$, resulting in a sparse matrix $\mathbf{W}'$. Let $X \subset \mathbb{R}^n$ be a finite input set of cardinality $N$, with each element having continuous entries bounded by $[c, d]$.

Then, by Lemma A.1, with probability

$$\gamma \geq 1 - \binom{N}{2} \rho^{k \, m},$$

the map $f = \sigma(\mathbf{W}' \cdot)$ is injective on $X$, where $k \geq 1$ is the minimum number of non-zero components in $\mathbf{x}_u - \mathbf{x}_v$ for all distinct pairs $\mathbf{x}_u, \mathbf{x}_v \in X$. We also recall from the argument in the proof of Theorem 3.2 that, for $\rho > 0$, $\gamma > 0$, and fixed $N, k$, one obtains a sufficient condition on $m$, namely $m \geq \log_\rho \left((1 - \gamma)\binom{N}{2}^{-1}\right) k^{-1}$

We now extend Lemma A.1 from injectivity to *non-colinearity* of the outputs. Specifically, we wish to show that for any two *distinct* inputs $\mathbf{x}_u, \mathbf{x}_v \in X$, the probability that there exists a nonzero scalar $\eta \in \mathbb{R}$ with

$$f(\mathbf{x}_u) = \eta f(\mathbf{x}_v)$$

is negligible (indeed measure zero) under our assumptions on the continuous distribution of weights and the independent pruning process.

We proceed as follows. Fix a single pair of distinct inputs $\mathbf{x}_u \neq \mathbf{x}_v \in X$. Denote by $\mathbf{w}'_r$ the $r$-th row of $\mathbf{W}'$. Then

$$f(\mathbf{x}_u) = \eta f(\mathbf{x}_v) \quad \Longleftrightarrow \quad \sigma(\mathbf{w}'^T_r \mathbf{x}_u) = \eta \sigma(\mathbf{w}'^T_r \mathbf{x}_v) \quad \text{for all } r = 1, \dots, m.$$

Because $\sigma$ is injective (elementwise), each equation

$$\sigma(\mathbf{w}'^T_r \mathbf{x}_u) = \eta \sigma(\mathbf{w}'^T_r \mathbf{x}_v)$$

for a *fixed* $\eta$ imposes a lower-dimensional (measure-zero) constraint on the row $\mathbf{w}'_r$ in $\mathbb{R}^n$. Enforcing this condition *across all* $r = 1, \dots, m$ and with the *same* scalar $\eta$ yields an intersection of these measure-zero sets in $\mathbb{R}^{m \times n}$. Consequently, the probability that

$$\mathbf{W}' \in \Big\{\mathbf{W}' \in \mathbb{R}^{m \times n} : f(\mathbf{x}_u) = \eta f(\mathbf{x}_v)\Big\}$$

is itself zero under our continuous distribution for $\mathbf{W}$ and subsequent Bernoulli pruning of entries (c.f. Lemma A.1). The only way the dimension of these constraints would fail to be negligible is if $\mathbf{W}'$ had some degenerate structure (e.g. all-zero rows, or precisely tuned rows to force colinearity), but for almost all choices of non-pruned weights in a sufficiently

overparameterized $\mathbf{W}'$ (i.e. with $m \geq \log_\rho \left( (1 - \gamma)\binom{N}{2}^{-1} \right) k^{-1}$ for $\gamma \to 1$), such a degeneracy cannot occur with non-zero probability.

Hence, for any *fixed* pair $\mathbf{x}_u, \mathbf{x}_v \in X$, the probability that they become mapped to colinear outputs under $f$ is zero. A union bound (Boole's inequality) over the $\binom{N}{2}$ distinct pairs in $X$ remains a finite union of measure-zero events. Therefore, with probability 1, *none* of the pairs $\mathbf{x}_u, \mathbf{x}_v \in X$ yield colinear outputs.

Putting it all together:

- *Injectivity*: we already know from Lemma A.1 that $P[f(\mathbf{x}_u) \neq f(\mathbf{x}_v) \forall \mathbf{x}_u \neq \mathbf{x}_v] \geq 1 - \binom{N}{2}\rho^{km}$.

- *Non-colinearity*: with probability 1, no two distinct inputs map to outputs that are scalar multiples of each other.

Since a measure-zero event does not further reduce the probability threshold from the injectivity part, the second property is effectively guaranteed *almost surely* in our random draw of $\mathbf{W}'$. Therefore, for any two distinct points in $X$, both

$$f(\mathbf{x}_u) \neq f(\mathbf{x}_v) \quad \text{and} \quad f(\mathbf{x}_u) \not\parallel f(\mathbf{x}_v)$$

hold with high probability (and indeed the colinearity condition holds with probability 0).

Thus, if $m$ is chosen large enough to satisfy $m \geq \log_\rho \left( (1 - \gamma)\binom{N}{2}^{-1} \right) k^{-1}$ then $\gamma \geq 1 - \binom{N}{2}\rho^{km} > 0$, ensuring both injectivity and non-colinearity of the outputs as claimed. $\qquad\square$

### A.4. Proof of Lemma 3.5

*Proof.* Let $G_1$ and $G_2$ be two graphs with adjacency and features matrices $\mathbf{A}_1 = \mathbf{A}_2$ and $\mathbf{X}_1 \neq \mathbf{X}_2$, respectively. Let $\mathbf{W}^{(1)}$ be the first weight's matrix of the first MP layers MLP and $\mathbf{M}^{(1)}$ it's associated pruning mask. Suppose $\mathbf{A}_1\mathbf{X}_1\mathbf{M}^{(1)} = \mathbf{A}_2\mathbf{X}_2\mathbf{M}^{(1)}$. Then, we need to show that for any weight matrix $\mathbf{W}^{(1)}$, $\mathbf{A}_1\mathbf{X}_1\mathbf{W}^{(1)} = \mathbf{A}_2\mathbf{X}_2\mathbf{W}^{(1)}$.

Now, assume, for contradiction, that there exists a weight matrix $\mathbf{W}^{(1)}$ such that $\mathbf{A}_1\mathbf{X}_1\mathbf{W}^{(1)} \neq \mathbf{A}_2\mathbf{X}_2\mathbf{W}^{(1)}$. Consider the elementwise product $\mathbf{M}^{(1)} \odot \tilde{\mathbf{A}}^{(1)} \odot \mathbf{W}^{(1)}$. Since $\mathbf{M}^{(1)}$ and $\tilde{\mathbf{A}}^{(1)}$ are binary matrices, this product selects certain entries of $\mathbf{W}^{(1)}$.

If $\mathbf{A}_1\mathbf{X}_1\mathbf{W}^{(1)} \neq \mathbf{A}_2\mathbf{X}_2\mathbf{W}^{(1)}$, there must exist some $i, j$ for which

$$\langle (\mathbf{A}_1\mathbf{X}_1)_{i,:}, \mathbf{W}^{(1)}_{:,j} \rangle \neq \langle (\mathbf{A}_2\mathbf{X}_2)_{i,:}, \mathbf{W}^{(1)}_{:,j} \rangle.$$

Since we can focus on the entries selected by $\mathbf{M}^{(1)} \odot \tilde{\mathbf{A}}^{(1)}$, we consider

$$\langle (\mathbf{A}_1\mathbf{X}_1)_{i,:}, (\mathbf{M}^{(1)} \odot \tilde{\mathbf{A}}^{(1)} \odot \mathbf{W}^{(1)})_{:,j} \rangle \quad \text{and} \quad \langle (\mathbf{A}_2\mathbf{X}_2)_{i,:}, (\mathbf{M}^{(1)} \odot \tilde{\mathbf{A}}^{(1)} \odot \mathbf{W}^{(1)})_{:,j} \rangle.$$

If these two inner products differ, then there exists at least one $l, l'$ such that

$$(\mathbf{A}_1\mathbf{X}_1)_{i,l}(\mathbf{M}^{(1)}_{l,j}\tilde{\mathbf{A}}^{(1)}_{l,j}\mathbf{W}^{(1)}_{l,j}) \neq (\mathbf{A}_2\mathbf{X}_2)_{i,l'}(\mathbf{M}^{(1)}_{l',j}\tilde{\mathbf{A}}^{(1)}_{l',j}\mathbf{W}^{(1)}_{l',j}).$$

For this difference to depend on $\mathbf{W}^{(1)}$, the corresponding entries of $\mathbf{M}^{(1)}$ and $\tilde{\mathbf{A}}^{(1)}$ must be nonzero. In particular, if there is a discrepancy when using $\mathbf{W}^{(1)}$, then choosing $\mathbf{W}^{(1)}$ such that it matches the nonzero structure selected by $\mathbf{M}^{(1)}$ and $\tilde{\mathbf{A}}^{(1)}$ would produce the same discrepancy. Hence, a difference in $\mathbf{A}_1\mathbf{X}_1\mathbf{W}^{(1)}$ and $\mathbf{A}_2\mathbf{X}_2\mathbf{W}^{(1)}$ would imply a difference in $\mathbf{A}_1\mathbf{X}_1\mathbf{M}^{(1)}$ and $\mathbf{A}_2\mathbf{X}_2\mathbf{M}^{(1)}$, contradicting the initial assumption.

Thus, our contradiction shows that no such $\mathbf{W}^{(1)}$ can exist. Therefore, for any $\mathbf{W}^{(1)}$, it must hold that $\mathbf{A}_1\mathbf{X}_1\mathbf{W}^{(1)} = \mathbf{A}_2\mathbf{X}_2\mathbf{W}^{(1)}$. $\qquad\square$

### A.5. Proof of Lemma 3.6

*Proof.* Under the assumption that classes are uniformly represented, each of the $C$ classes contains about $\frac{M}{C}$ of these $M$ indistinguishable graphs, and the pairwise indistinguishability of isomorphism types of different classes is equivalent to

assigning all $M$ graphs to a single class. Then, the model correctly classifies exactly those $\frac{M}{C}$ graphs that truly belong to the chosen class. The other $M - \frac{M}{C} = M\left(1 - \frac{1}{C}\right)$ graphs from the remaining $C - 1$ classes are misclassified.

For the remaining $N - M$ graphs, which are all distinguishable isomorphism types, the model can correctly classify all of them (assuming perfect separability in the distinguishable subset).

Hence, the total number of correctly classified graphs is

$$(N - M) + \frac{M}{C}.$$

Dividing by $N$ to obtain the accuracy:

$$\frac{(N - M) + \frac{M}{C}}{N} = 1 - \frac{M}{N} + \frac{M}{CN}.$$

Since $M \approx \frac{UN}{I}$, substitute this into the equation:

$$= 1 - \frac{U}{I} + \frac{U}{IC}.$$

Factor out $\frac{U}{I}$:

$$= 1 - \frac{U}{I}\left(1 - \frac{1}{C}\right).$$

This expression gives the maximum fraction of correctly classified graphs under the stated assumptions.

$\square$

## B. DATASETS

*Table 2.* Overview of selected datasets.

| DATASET | TYPE | #GRAPHS | AVG. NODES | AVG. EDGES | LABELS | TASK |
|---------|------|---------|-----------|-----------|--------|------|
| MUTAG | CHEMICAL | 188 | 17.9 | 19.8 | NODE, EDGE | CLASS. |
| AIDS | CHEMICAL | 2,000 | 15.7 | 16.2 | NODE, EDGE | CLASS. |
| PTC_FM | CHEMICAL | 349 | 14.1 | 14.5 | NODE, EDGE | CLASS. |
| PTC_MR | CHEMICAL | 344 | 14.3 | 14.7 | NODE, EDGE | CLASS. |
| NCI1 | CHEMICAL | 4,110 | 29.9 | 32.3 | NODE | CLASS. |
| PROTEINS | PROTEIN | 1,113 | 39.1 | 72.8 | NODE | CLASS. |
| ENZYMES | PROTEIN | 600 | 32.6 | 62.1 | NODE | CLASS. |
| MSRC_9 | IMAGE | 221 | 40.6 | 72.4 | NODE | CLASS. |
| MSRC_21C | IMAGE | 563 | 77.5 | 142.8 | NODE | CLASS. |
| IMDB-BINARY | SOCIAL | 1,000 | 19.8 | 96.5 | - | CLASS. |

To examine the complex relationship between the Lottery Ticket Hypothesis (LTH) and the expressivity of Graph Neural Networks (GNNs), we utilize a diverse set of ten real-world datasets. Each dataset is carefully selected based on its relevance to distinct graph structures and domain-specific tasks. These datasets are sourced from the widely recognized TUDataset collection (Morris et al., 2020), which serves as a standard benchmark for tasks involving graph classification and regression. A detailed summary of these datasets is presented in Table 2.

**Chemical Compounds:** The MUTAG, AIDS, PTC_FM, PTC_MR, and NCI1 datasets focus on chemical compounds, where molecular structures are represented as graphs, with nodes corresponding to atoms and edges denoting chemical bonds. MUTAG, one of the earliest and most widely used datasets for graph classification, comprises nitroaromatic compounds labeled by their mutagenic effects on Salmonella typhimurium. The AIDS dataset contains molecular graphs relevant to anti-HIV drug discovery, aiming to predict inhibitory activity against HIV. The PTC datasets (FM and MR) involve compounds evaluated for rodent carcinogenicity, classified based on different experimental setups. NCI1, a larger dataset derived from the National Cancer Institute's screening program, involves classifying compounds according to their activity against non-small cell lung cancer.

**Protein Structures:** The PROTEINS and ENZYMES datasets are derived from bioinformatics, where graphs are used to represent protein structures. In the PROTEINS dataset, nodes correspond to secondary structural elements, such as helices and sheets, with edges reflecting spatial proximity. The classification task is to determine whether a given protein functions as an enzyme. The ENZYMES dataset builds on this by categorizing enzymes into one of the six top-level classes defined by the Enzyme Commission (EC), based on the types of chemical reactions they catalyze (Borgwardt et al., 2005; Schomburg et al., 2002).

**Image Segmentation:** The MSRC_9 and MSRC_21C datasets, originating from the MSRC database, are designed for semantic image segmentation tasks. In these datasets, images are represented as graphs, where nodes correspond to superpixels and edges capture spatial relationships between them. The task requires GNNs to classify nodes into various categories based on the visual content of the superpixels. MSRC_21C serves as an extended version, offering a greater number of classes and increased complexity (Neumann et al., 2016).

**Social Networks:** The IMDB-BINARY dataset represents social networks, with each graph modeling the collaboration network of actors who have appeared together in movies. The classification task involves distinguishing these networks based on the movie genre, specifically differentiating between action and romance. This dataset highlights the complexities of real-world social networks, where nodes correspond to individuals and edges represent their interactions, challenging GNNs to capture nuanced social dynamics (Yanardag & Vishwanathan, 2015).

These datasets collectively offer a diverse evaluation framework, encompassing a wide variety of graph structures and complexities, ranging from small molecular graphs to larger social networks and image-based graphs. Leveraging this extensive set of benchmarks allows our empirical analysis to thoroughly evaluate the hypothesis across multiple domains.

