# OpenReview forum: "Weisfeiler and Leman Go Gambling: Why Expressive Lottery Tickets Win"
_ICML.cc/2025/Conference — ICML 2025 poster_

### Official Review · Reviewer_wwh4 · 2025-03-13

**Overall Recommendation:** 4

**Summary:**

The paper explores the connection between the Weisfeiler-Lehman (WL) test and the Lottery Ticket Hypothesis (LTH). The authors establish criteria for pruning mechanisms, requiring that the pruned network remain as expressive as the original in terms of the WL test to preserve its performance. They define the concept of "critical paths"—key weight subsets essential for distinguishing non-isomorphic graphs—and demonstrate that there exists a subset of weights in the original network which will distinguish the same set of graphs as the original network. Additionally, they analyze how selecting suboptimal "lottery tickets" (in terms of expressivity) impacts classification accuracy, both theoretically and empirically.

#### update
I am satisfied with the author response to my questions. I think this is an interesting piece of work so I maintain my score.

**Claims And Evidence:**

All claims are supported by clear and convincing evidence.

**Essential References Not Discussed:**

I am not aware of any essential references which are not discussed.

**Experimental Designs Or Analyses:**

Overall, the experiments are quite thorough -- the authors test on standard graph benchmarks. However, I think one could further strengthen the experiments by also considering attention-based graph neural networks such as GAT. I would also be curious to see if the results hold for "less expressive" GNN architectures which use max/min-aggregation. Finally, the authors use random pruning but I wonder if the authors have thought of any heuristics for which we can use for pruning which intuitively align with keeping critical paths in the network?

**Methods And Evaluation Criteria:**

The benchmarks make sense for the problem.

**Other Comments Or Suggestions:**

I believe that the authors mainly consider classification tasks in their paper (esp. re: Lemma 3.6 which considers how the quality of lottery tickets affects accuracy) -- it wasn't clear to me at first, so maybe they can mention that more clearly in their contributions.

There were also several places where I got a bit confused with the notation, maybe the authors can add some clarifications:

(1) on line 187-190, the authors start using $\hat{D}$ and $\hat{\phi}$. I assume that this means a pruned set of graphs and a pruned network but I didn't see this defined previously.

(2) In Lemma 3.6, they say $U \leq I$ and I guess that U is the set of graph types which are not distinguishable by the model in question but perhaps they can say this more clearly.

**Other Strengths And Weaknesses:**

Strengths: To the best of my knowledge, the link between the lottery ticket hypothesis to the Weisfeiler-Lehman test (a common way for people measure the expressivity of GNNs) has not been explored so this paper represents an exciting contribution linking GNN expressivity with LTH. The theoretical contributions are also well-supported by the experiments and provide some theoretical insight as to how practitioners should prune networks and what kind of features they might want to preserve during pruning.

Weaknesses: I already made a comment above regarding the experiments but I'll also make a comment here -- I think the experiments (while already extensive) could be further improved by considering more network varieties than just GIN and GCN. In particular, I would be interested to see if their results would hold empirically for less expressive architectures (such as max/min-aggregation GNNs) i.e. we see similar trends as GCN and GIN when we prune out weights for max/min aggregation networks or maybe because these networks are inherently less expressive, the drop in accuracy would not be as dramatic?

**Questions For Authors:**

I have no questions which will change my evaluation of the paper.

**Relation To Broader Scientific Literature:**

Both the lottery ticket hypothesis and the Weisfeiler-Lehman test are well-studied in previous literature. In particular, [XHLJ '19] explicitly connect the WL-test to GNN expressivity and argue that any GNN with an injective aggregation and readout function will have the same expressivity as the WL test. Additionally, to the best of my knowledge, graph LTH was introduced in  [CSCZW '21] but it seems that most work in graph LTH is mostly empirical work which is focused on different pruning techniques. This paper connects GNN expressivity with LTH research and establishes conditions under which the expressivity of a trained network will be maintained even after pruning.

"How powerful are Graph Neural Networks" [XHLJ '19]
 "A unified lottery ticket hypothesis for graph neural networks" [CSCZW '21]

**Theoretical Claims:**

I checked the proof for Theorem 3.2 and I did not find any issues.

---

> ### Author Rebuttal · Authors · 2025-03-31
>
> We thank you for your review and hope our response below satisfactorily addresses your questions.
>
> > However, I think one could further strengthen the experiments by also considering attention-based graph neural networks such as GAT.
>
> Our theoretical results apply to general moment-based GNN architectures and we thus expect the formal insights we develop -- e.g., the connection between pruning, critical path removal, and loss of expressivity (e.g., Theorem 3.2, Lemma 3.5) -- to apply broadly across this architectural class, which includes GAT as well. As such, our upper bounds on achievable classification accuracy under misaligned pruning (e.g., Lemma 3.6) also hold. We emphasize that these bounds are theoretical and are meant to illustrate structural limitations.
>
> However, refining our formal analysis to architectures beyond this most general setting, (including the effects of attention or edge features that modulate aggregation) is a promising direction for future work which might reveal additional, architecture-specific vulnerabilities not covered by our existing work.
>
> We acknowledge that an empirical evaluation would underpin any theoretical deliberations concerning GAT and LTH.
>
> > Finally, the authors use random pruning but I wonder if the authors have thought of any heuristics for which we can use for pruning which intuitively align with keeping critical paths in the network?
>
> As stated in our response to Reviewer 3s8a, an iterative pruning mechanism that enforces injectivity of each transformation on its local input domain would be maximally expressive on a given dataset, thus preserving the paths needed for the GNN to distinguish non-isomorphic graphs associated with different classes.
>
> > I believe that the authors mainly consider classification tasks in their paper (esp. re: Lemma 3.6 which considers how the quality of lottery tickets affects accuracy) -- it wasn't clear to me at first, so maybe they can mention that more clearly in their contributions.
>
> This is correct—we consider the graph classification setting. However, with minor adjustments, our results could likely also be transferred to node classification scenarios. We will clarify this in the final version of the paper.
>
> > In particular, I would be interested to see if their results would hold empirically for less expressive architectures (such as max/min-aggregation GNNs) i.e. we see similar trends as GCN and GIN when we prune out weights for max/min aggregation networks or maybe because these networks are inherently less expressive, the drop in accuracy would not be as dramatic?
>
> This is an interesting thought. The impact of pre-training pruning on expressivity might be less dramatic because there is simply less expressivity to lose. However, this also implies that the benefits of preserving expressivity (such as faster convergence and improved generalization) might not be as pronounced for these architectures.
>
> We set up experiments to test your hypothesis for both min and max aggregation in GIN/GCN. Preliminary results indicate that a) at least for max aggregation and low pruning percentages (< 50%), the accuracy drop with respect to an unpruned model also using max aggregation appears to be less pronounced than for add aggregation and b) that the probability that a given sparse initialization is a winning ticket seems to be less dependent on the model (i.e. GCN or GIN), which make sense, considering that using max aggregation likely reduces the differences in expressivity of GCN and GIN in general. The overall trend and relation to expressivity we observe, however, appears to be comparable to the one we observed for GIN/GCN with add aggregation.
>
> Given the timeframe of the rebuttal & discussion and the runtime of our setup (~8 weeks, compare line 352), we can likely only provide preliminary results, which might not be statistically reliable. We will consider including the full results for min/max aggregation in the final version of the paper.
>
> > (1) on line 187-190, the authors start using $\widehat{D}$ and $\widehat{\Phi}$. I assume that this means a pruned set of graphs and a pruned network but I didn't see this defined previously.
>
> Indeed, in lines 187-190  $\widehat{D}$ and $\widehat{\Phi}$ refer to pruned models or a dataset of pruned graphs, but we used the more general term “modified” there, as Criterion 1 is not only specific to pruning (which, at least for graphs, can take multiple forms—such as adjacency matrix pruning or node/edge feature pruning) but could potentially also be extended to, for example, quantization.
>
> > (2) In Lemma 3.6, they say and $U \leq I$ guess that $U$ is the set of graph types which are not distinguishable by the model in question but perhaps they can say this more clearly.
>
> $U$ in Lemma 3.6 represents the number of isomorphism types of dataset $D$ which are indistinguishable from at least one other isomorphism type present in the dataset. We will clarify this in the final version of the paper.

---

### Official Review · Reviewer_3s8a · 2025-03-14

**Overall Recommendation:** 4

**Summary:**

This paper deals with the Strong Lottery Ticket Hypothesis (SLTH) in the context of graph neural network (GNN).
Particularly, the authors argue that there exists an initialized GNN with sufficiently high expressivity that can match the original performance after training.
To demonstrate this, the authors theoretically show that there exists a sparse initialized GNN that matches 1-WL expressivity and the expressive GNN can generalized comparably to the dense one.
The experiments demonstrate that the more expressive a sparse initialized GNN is, the better the post-training network performs, supporting the theoretical analysis.

## Update after rebuttal

The authors' response has effectively addressed the raised questions, including a way to find an expressive GNN and a way to find a GLT in the context of expressivity. Thus, the reviewer has decided to maintain the original rating, 'Accept'.

**Claims And Evidence:**

The authors claim that a sparse initialized GNN exists with maximally expressive paths, and such a network potentially generalizes comparably to a dense network.
The proposed theoretical and empirical evidence effectively support the authors' claim.

**Essential References Not Discussed:**

None.

**Experimental Designs Or Analyses:**

For experiments, the authors focus on investigating the relationship between the expressivity of an initialized network and the performance (or expressivity) of a trained network.
The plot in Figure 4 and the Pearson correlation in Figure 3 clearly demonstrate the relationship.

**Methods And Evaluation Criteria:**

The authors do not propose a method and investigate the relationship between the expressivity of an initialized network and the performance (or expressivity) of a trained network.
In Figure 4, the authors present this relationship across various pruning ratios and datasets, and in Figure 3, the authors adopt Pearson correlation to evaluate the relationship.

**Other Comments Or Suggestions:**

How can we find a sparse GNN with significant expressivity?
The existing methods on the graph lottery ticket (GLT) effectively can find such a subnetwork?
Some suggestions for finding GLT and analyzing the previous GLT method in the context of expressivity would make this paper more interesting.

**Other Strengths And Weaknesses:**

All are mentioned in other sections.

**Questions For Authors:**

In GNN, the authors analyze the lottery ticket hypothesis from the lens of expressivity.
Then, I’d like to ask what the analogous concept of expressivity is in the context of other neural networks, such as CNNs or MLPs?

**Relation To Broader Scientific Literature:**

As someone unfamiliar with GNNs, I find that the authors effectively introduce the necessity of revealing the key to GNN generalization.
It is interesting to see the connection between pre-training expressivity and post-training generalization, and I believe this observation makes a significant contribution to the literature.

**Theoretical Claims:**

In appendix A, the authors prove the existence of maximally expressive paths within an initialized GNN and the trainability of a sparse network with the paths.
The proofs are well derived, and I did not find any significant issues.

---

> ### Author Rebuttal · Authors · 2025-03-31
>
> Thank you for your thoughtful review. We hope that our response below addresses your questions satisfactorily.
>
> > How can we find a sparse GNN with significant expressivity?
>
> For moment-based architectures (see line 126 right column, Lemma 2.6), such as those analyzed in our work, maximal expressivity (i.e., WL-equivalence for GIN) is achieved if all aggregation and combination functions are injective over their respective domains defined by the dataset and prior operations. A simple approach to expressive pre-training sparsification is to begin at the first message-passing layer and, after initialization, for a given pruning percentage, sample $k$ configurations of that layer and select the one for which the number of unique input node vectors equals the number of unique output node vectors over all graphs. This procedure is then repeated for subsequent layers, increasing the pruning ratio until no injective configuration is found within a computationally feasible sample size $k$. The resulting network, with guaranteed injective transformations, is sparsified yet maximally expressive. We will develop and analyze more sophisticated techniques in future work.
>
> > The existing methods on the graph lottery ticket (GLT) effectively can find such a subnetwork?
>
> Existing methods typically employ iterative and/or gradient-based approaches to prune graphs or transformations in GNNs. In many cases, expressivity is either omitted entirely (e.g., [1, 2, 3, 4]) or mentioned in a side note as an important concept without further formal analysis or empirical validation (e.g., [5]). To the best of our knowledge, we are the first to provide a clear formal analysis connecting the Lottery Ticket Hypothesis for GNNs with expressivity, defined strictly as the ability to distinguish non-isomorphic graphs.
>
> > Some suggestions for finding GLT and analyzing the previous GLT method in the context of expressivity would make this paper more interesting.
>
> We do not search graph lottery tickets (GLTs), which prune the input graphs, but focus on the effect of pruning the parameters of the learnable transformations of a GNN on its expressivity. We consider developing practical, expressivity-oriented pruning methods based on the theoretical insights of our work and the comparison to existing pruning methods a highly relevant topic for future research that we definitely want to explore further.
>
> > In GNN, the authors analyze the lottery ticket hypothesis from the lens of expressivity. Then, I’d like to ask what the analogous concept of expressivity is in the context of other neural networks, such as CNNs or MLPs?
>
> In the context of CNNs and MLPs, the analogous term typically used is expressive power. For MLPs, expressive power refers to the class of functions that can be approximated under architectural constraints (width, depth, activation), while for CNNs, it describes the types of features extracted by convolutional filters. However, the definition remains vague for CNNs and MLPs, lacking a baseline comparison like the WL test and its variants in GNNs.
>
> [1] T. Chen et. al, A Unified Lottery Ticket Hypothesis for Graph Neural Networks, ICML, 2021
>
> [2] B. Hui et. al, Rethinking Graph Lottery Tickets: Graph Sparsity Matters, ICLR, 2023
>
> [3] A. Tsitsulin, The Graph Lottery Ticket Hypothesis: Finding Sparse, Informative Graph Structure, CoRR/abs.2312.04762, 2023
>
> [4] Y.D. Sui et al., Inductive Lottery Ticket Learning for Graph Neural Networks, Journal of COmputer Science and Technology, 2023
>
> [5] K. Wang et al., Searching Lottery Tickets in Graph Neural Networks: A Dual Perspective, ICLR, 2023

---

### Official Review · Reviewer_3Ejh · 2025-03-15

**Overall Recommendation:** 3

**Summary:**

This paper studies the role of Graph Neural Network (GNN) expressivity in Lottery Ticket Hypothesis (LTH), in particular, the conditions NN pruning mechanisms must satisfy to maintain prediction quality. They show that trainable sparse subnetworks exist within moment-based GNNs, matching 1-WL expressivity. They also show the importance of preserving critical computational paths to prevent performance degradation. The key claim is that expressive sparse initializations improve generalization and convergence, while improper pruning can lead to irrecoverable expressivity loss.

**Claims And Evidence:**

Mostly yes. A few problematic claims are as follows:
1. Abstract says that - “… and subsequently show that an increased expressivity in the initialization potentially accelerates model convergence and improves generalization.” => I did not find any explicit, theoretical or empirical results that support this claim. Theorem 3.3 is not conclusive in this regard. It merely suggests the following, as per the authors,  “.. a model initialized such that two graphs do not receive (partially) identical node embeddings is likely to converge faster and generalize more effectively and thus more likely to be a winning ticket in the initialization lottery, as identical embeddings are always codirectional.” To my understanding, an empirical study is lacking in the paper that directly validates this statement.
2. Section 1.2 says -“We formally link GNN expressivity to LTH by establishing criteria that pruning mechanisms—both graph and parameter pruning must satisfy to preserve prediction quality.” => However, if "graph pruning" refers to pruning the adjacency matrix structure, no explicit criteria are provided to ensure the preservation of prediction quality. The absence of a rigorous theoretical or experimental analysis for graph pruning makes this claim problematic.

**Essential References Not Discussed:**

None, to the best of my knowledge.

**Experimental Designs Or Analyses:**

The experiments and Results section seemed rushed lacking care and clarity. A few questions/suggestions include:
1. There are instances such as lines 352-358, where one sentence spans 6-7 lines. Consider the first paragraph in Section 5 as another example. Please simplify such sentences to improve readability.
2. Please explain the meaning of $\vartheta$, $S$ and the term “clean accuracy” mentioned in the caption of Figure 2.
3. Please discuss how you compute the probability $P(WT(GNN) | \rho , \tau_{pre}, \epsilon)$ in Figure 2.
4. Mention which dataset has been used to generate Figure 2.
5. Figure 4 is not an ideal way to represent the result, there are too many data points to understand what story they tell. \
6. Lines 409-412 says: “Moreover, we find that a GNN, when initialized with certain sparsity level and non-zero weights trained in isolation, is highly unlikely to transition to a higher expressivity state (Table 1).” => Based on which column(s) in Table 1 did you draw this conclusion, and how?

**Methods And Evaluation Criteria:**

1. The datasets make sense as they are well-known benchmarks for graph classification tasks.

2. However, the evaluation criteria appear to be vague and under-explained in Section 4. In particular, I urge the authors to please explain, with examples, (a) how they measure expressivity $\tau$, and (b) how they measure whether two embeddings are distinguishable or not.

**Other Comments Or Suggestions:**

1. In Section 2, line 114 is incorrect. It should be => $(\phi(u),\phi(v))$ in $E(H)$ for all $(u,v)$ in $E(G)$
2. Define $\Sigma$ in line 120 where it is used for the first time.
3. Line 120, “A function $V(G) \rightarrow \Sigma$”  => A function $l: V(G) \rightarrow \Sigma$.
4. The notations in section 3 could be improved and made more unambiguous.

**Other Strengths And Weaknesses:**

The study is original and significant. However many parts of the paper lack clarity and readability. Some claims are problematic and some experiments were not clearly explained. Some other issues include:
1. Issues with Lemma 3.6 => The conditions in Lemma 3.6 appears to be unrealistic, for instance, how could  someone know the  isomorphism types $I$ of a given dataset apriori? Can you empirically validate Lemma 3.6 with some of your datasets?
2. See the Experimental Designs Or Analyses section for details on experiment-related issues.
3. See the Methods And Evaluation Criteria section for evaluation-related issues
4. See the Claims And Evidence section for claim-related issues.

**Questions For Authors:**

Could you please clarify the issues mentioned in the Claims And Evidence section, Experimental Designs Or Analyses section, the Methods And Evaluation Criteria section, and Other Strengths And Weaknesses section?

**Relation To Broader Scientific Literature:**

Relevant to the GNN community.

**Theoretical Claims:**

I have not carefully checked the correctness of the proofs, hence cannot comment upon their correctness.

---

> ### Author Rebuttal · Authors · 2025-03-31
>
> We appreciate your detailed review. Below, we address your concerns and will incorporate the corresponding changes into the final version. We are looking forward to further discussions with you.
>
> > Abstract says that [...]
>
> We acknowledge Theorem 3.3 does not directly guarantee improved convergence or generalization but links embedding distinctiveness to gradient diversity, a known factor influencing both -- hence our cautious use of “potentially” in the abstract. Empirically (Section 5), since all models were trained for the same number of epochs, the consistently superior performance of sparse initializations with high expressivity suggests improved convergence and generalization, consistent with our theory.
>
> > Section 1.2 says [...]
>
> With "graph pruning", we refer to the removal of information from input graphs (i.e. pruning adjacency matrices, node features or edge features). Criterion 1 (Section 3) covers adjacency matrix pruning via modified tuples $\widehat{D}$ (e.g., $D_l = (\widehat{A}, X, t)$) and requires that non-isomorphic graphs from different classes remain distinguishable; otherwise, the maximal achievable accuracy degrades. Since we focus on parameter pruning, subsequent sections address only that case.
>
> > (a) how they measure expressivity
>
> As no standard method exists, we adopt an approach that balances efficiency and clarity.
> For expressivity measurement, we retain one representative per isomorphism type, as isomorphic graphs yield identical embeddings by GNN permutation invariance. Let $\{G_1,\ldots,G_n\}$ be the $n$ non-isomorphic graphs with embedding vectors $\{h_{G_1},\ldots,h_{G_n}\}$. We mark each pair $(h_{G_i}, h_{G_j})$ as indistinguishable if $h_{G_i} - h_{G_j} = 0$. The expressivity $\tau$ is the fraction that remains distinguishable. Alternative methods (e.g., exhaustive comparisons or training accuracy) are more costly or reflect different notions of expressivity. We chose our approach for its scalability (over 13,500 runs) and direct assessment of whether a non-isomorphic graph is distinguishable from the rest of the dataset.
>
> > (b) how they measure [...] distinguishable or not
>
> For efficiency, we assess distinguishability by checking whether $h_{G_1} - h_{G_2} \neq 0$ after summation-based readout of the final MP-layer outputs. This generally implies differing node-level embeddings. While distinct node embeddings can theoretically sum to the same vector, such collisions are extremely rare and occur only under measure-zero configurations.
>
> > 1. [...] one sentence spans 6-7 lines.
>
> We will revise the relevant sections to improve clarity and conciseness.
>
> > 2. [...] explain [...] $\vartheta$, S, and [..] “clean accuracy”
>
> The variables $\vartheta$ and $\varepsilon$ in Fig. 2 are used to group similar $\tau$ values into intervals, as observing an exact empirical value of $\tau$ is unlikely. The set $S$ used in computing $\overline{\Delta}$ contains models for which $\tau_{\mathrm{pre}} \in [\vartheta \pm \varepsilon]$. The term "clean accuracy" refers to the accuracy of the dense, unpruned model.
>
> > 3. [...] how you compute the probability in Fig. 2.
>
> We fix a target $\vartheta$ and collect runs with $\tau_{\mathrm{pre}} \in [\vartheta \pm \varepsilon]$. A run is labeled a “winning ticket” if its accuracy drops by less than $5$% compared to the unpruned model. The probability is the fraction of such runs, aggregated (with subset-size normalization) to plot winning ticket probabilities across pruning levels and thresholds.
>
> > 4. Mention which dataset [...] Fig. 2.
>
> Fig. 2 displays data generated from all runs and therefore all 10 datasets, which are listed in Tab. 2, Appendix B.
>
> > 5. Fig. 4 is not an ideal [...]
>
> As illustrated in our Fig.4, none of the data points fall in the upper left half, indicating that $\tau_{\mathrm{post}}$ is generally lower than $\tau_{\mathrm{pre}}$. This supports our claim that pruned models typically do not gain expressivity during training, underscoring the importance of expressive sparse initialization.
>
> > 6. Lines 409-412 says: [...]
>
> All columns of Tab. 1 support our claim. If a GNN initialized and trained as in lines 409–412 could reliably transition from low $\tau_{\mathrm{pre}}$ to high $\tau_{\mathrm{post}}$, then for some $\kappa$, this would occur with a probability above a low single-digit percentage. While we do not specify a threshold for high probability, Fig. 4 reflects the trend shown in Tab. 1.
>
> > Issues with Lemma 3.6 [...]
>
> Lemma 3.6 is a theoretical bound requiring knowledge of all isomorphism types, which is impractical—though tools like nauty can identify them, and many benchmarks include them. Most datasets also lack the assumed uniform class distribution. The lemma is meant to conceptually illustrate how misaligned pruning limits a GNN’s maximal accuracy. Refining it for more realistic settings is a promising direction for future work.

---

### Decision · Program_Chairs · 2025-05-01

**Decision:**

Accept (poster)

**Comment:**

This submission mainly focuses on the lottery tickets hypothesis in graphs. Although there are lots of existing studies in GNN LTH, the authors study the problem from a novel angle. They theoretically show that there exists a sparse initialized GNN that matches 1-WL expressivity, and the expressive GNN can generalize comparably to the dense one. They also show the importance of preserving critical computational paths to prevent performance degradation. Overall, we believe it is a strong paper and we recommend an acceptance.